# SketchingReality: From Freehand Scene Sketches to Photorealistic Images

**Ahmed Bourouis**[1]     **Mikhail Bessmeltsev**[2]     **Yulia Gryaditskaya**[1,3]

[1]University of Surrey, UK     [2]Université de Montréal, Canada     [3]Adobe Research, UK

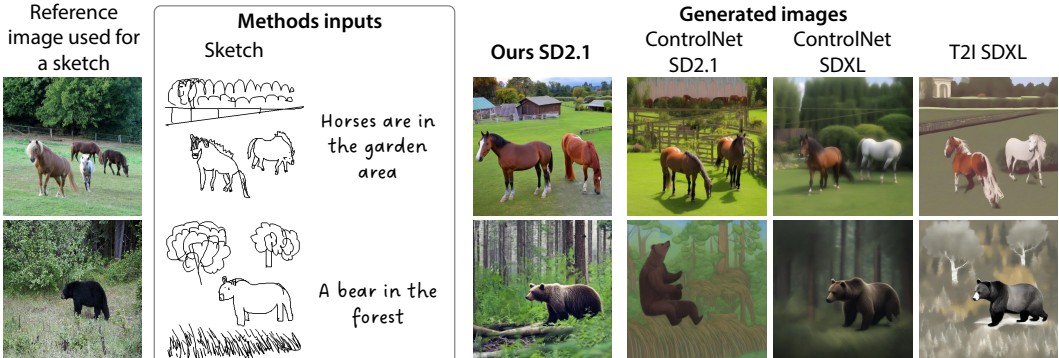

Figure 1: With just a few strokes, sketches can convey complex visual concepts that are difficult to express in words, making them a natural conditioning choice for efficient, human-centered, controllable generative AI. Our method takes as input a freehand sketch together with a text prompt. The figure compares our results with state-of-the-art baselines[1]: ControlNet (Zhang et al., 2023) and T2I-Adapter (Mou et al., 2023) — on freehand sketches from the FS-COCO dataset (Chowdhury et al., 2022). SD2.1 and SDXL represent the used backbones: Rombach et al. (2022) and Podell et al. (2023), respectively. The first column shows the reference image presented to participants, who recreated it from memory within a limited time, simulating how humans draw from a mental image. Our approach achieves a strong balance between sketch adherence and photorealism.

## Abstract

Recent years have witnessed remarkable progress in generative AI, with natural language emerging as the most common conditioning input. As underlying models grow more powerful, researchers are exploring increasingly diverse conditioning signals — such as depth maps, edge maps, camera parameters, and reference images — to give users finer control over generation. Among different modalities, sketches are a natural and long-standing form of human communication, enabling rapid expression of visual concepts. Previous literature has largely focused on edge maps, often misnamed "sketches", yet algorithms that effectively handle true freehand sketches — with their inherent abstraction and distortions — remain underexplored. We pursue the challenging goal of balancing photorealism with sketch adherence when generating images from freehand input. A key obstacle is the absence of ground-truth, pixel-aligned images: by their nature, freehand sketches do not have a single correct alignment. To address this, we propose a modulation-based approach that prioritizes semantic interpretation of the sketch over strict adherence to individual edge positions. We further introduce a novel loss that enables training on freehand sketches without requiring ground-truth pixel-aligned images. We show that our method outperforms existing approaches in both semantic alignment with freehand sketch inputs and in the realism and overall quality of the generated images. The code is available on our project webpage.

---

[1]For ControlNet and T2I-Adapter, we used parameter settings selected via a user study and qualitative evaluation, detailed in Sec. 4.4.1 and Sec. C.

# 1 INTRODUCTION

In recent years, generative AI for visual content such as images and videos has progressed dramatically. While research continues to push the boundaries of quality and realism, enabling greater user control has become increasingly important. Researchers are exploring diverse conditioning signals complementary to natural language — such as depth maps, edge maps, camera parameters, and reference images. In this work, we focus on human-drawn (freehand) sketches, which, alongside language, are among the oldest forms of human communication (Donald, 1993). With just a few strokes, sketches can convey complex visual concepts that are difficult to express in words, making them a natural conditioning choice for efficient, human-centered, controllable generative AI. Yet algorithms that effectively handle true freehand sketches — with their inherent abstraction and distortions — remain underexplored. Critically, we distinguish between edge maps, often regarded as "sketches" in the literature, and genuine freehand sketches. Perceptual studies show that freehand sketches frequently represent abstracted object forms (Tversky, 2002; Eitz et al., 2012) and may include distorted proportions or relative sizes (Hertzmann, 2025). Fig. 1 illustrates examples from the FS-COCO dataset of scene sketches (Chowdhury et al., 2022). The first column shows the reference image presented to the participants, who were then asked to recreate it from memory within a limited time, simulating how humans draw from a mental image. Objects in these sketches often exhibit high levels of abstraction — for example, in the second row of Fig. 1, grass is represented by a single strip of vertical strokes, while a forest is depicted with a few schematically drawn trees. Relative sizes of objects also frequently deviate substantially from the reference images.

While freehand sketches provide rich semantic cues, their abstraction, distortions, and ambiguity make them challenging for existing generative models to interpret effectively. Existing conditioning mechanisms often fail to capture the intended objects and relationships, producing images that either ignore important sketch details or compromise realism (Fig. 1). These limitations highlight two key challenges for generative models: (i) extracting meaningful semantic representations from highly abstract sketches, and (ii) generating images that are both realistic and respect the sketch. In our work, we pursue the challenging goal of balancing photorealism with sketch adherence when generating images from freehand input.

First, we observe that for freehand sketches, the image quality of typical conditioning mechanisms, such as ControlNet (Zhang et al., 2023) and Adapters (Mou et al., 2023), is limited by the latent features of their conditional input encoders. Specifically, the VAE encoder used in ControlNet-based models may lack semantic understanding of sketches, while the convolutional encoders in Adapter-based approaches may not be expressive enough to learn robust semantic representations. For the first time, we exploit semantic features from a CLIP-based sketch encoder (Bourouis et al., 2024), fine-tuning its final layers for our task. To inject richer semantic features, we propose a dedicated modulation network (Fig. 2a). This enables the encoder to maintain a semantically meaningful latent space while capturing finer-grained visual details from sketches.

Second, we propose a novel loss that sidesteps the need for pixel-aligned ground-truth images during training while preserving the semantics of the sketch input. To this end, we leverage the aforementioned semantic sketch encoder to estimate the likelihood of each sketch pixel belonging to a specific object category. During training, these semantic likelihoods guide the cross-attention between the corresponding textual tokens and the latent image representation, encouraging the generated image to follow the freehand sketch (Fig. 2b). We train on a mix of synthetic sketches, algorithmically generated from reference images, and freehand sketches. This approach enables the preservation of semantic information, important details, and the generation of natural-looking images.

As sketches are often ambiguous — for example, the trees in Fig. 1 (first row) are depicted as a set of arcs — we rely on text as an additional input. Fig. 1 (Ours SD2.1) demonstrates that our method strikes a strong balance between the realism of generated images and adherence to the sketch, preserving relative positions, orientations, and some finer details of objects, such as a bear facing left and the curve of its back (Fig. 1, second row). We validate our designs via qualitative and quantitative comparisons with previous methods, as well as via perceptual user studies.

In summary, our contributions are: (i) a new method that generates realistic images from freehand and potentially abstract scene sketches; (ii) the use of semantic sketch features through integration with the proposed modulation network in latent space; (iii) enabling efficient training on freehand sketches with a loss function that emphasizes the semantic structure of sketch inputs.

## 2    RELATED WORK

Although earlier work on a general sketch- or edge-based image generation relied on GAN-based approaches (Isola et al., 2017; Chen & Hays, 2018; Huang et al., 2018; Liu et al., 2020; Wang et al., 2021), recent work has demonstrated the superiority of diffusion-based approaches. In what follows, we focus our discussion on the latter. For a broader overview of spatial control in diffusion models, we refer the reader to recent surveys (Cao et al., 2024; Jiang et al., 2024).

### 2.1    PIXEL ALIGNED

Earlier work enabling control via sketches often considers precise, pixel-aligned edge maps. Some methods incorporate sketch conditions by concatenating them along the channel dimension, either in the pixel space (Cheng et al., 2023) or in the feature space (Huang et al., 2023; Unlu et al., 2024). These approaches often produce high-quality, pixel-aligned results, but typically involve training or at least fine-tuning the *entire* model. To avoid the overhead of training full models, several methods introduce sketch guidance through lightweight stand-alone modules integrated into pre-trained diffusion models. ControlNet (Zhang et al., 2023) does this via a dedicated branch with zero convolutions, significantly reducing the number of trainable parameters. T2I-Adapter (Mou et al., 2023) trains a lightweight convolutional encoder aligned with a U-Net denoiser. ControlNext (Peng et al., 2024) uses a lightweight encoder to learn conditional features and normalizes them to match the distribution of denoising features, improving convergence. Our work is most closely related to the UNet-based encoder–decoder method proposed by Ham et al. (2023), which modulates text-conditioned latents with sketches represented as three-channel RGB images and encoded by a UNet. Voynov et al. (2023) refine noisy latents using gradients of a similarity loss between edge-guidance latents and the output of a trained MLP applied to multi-level U-Net features. However, we show that none of these methods generalizes reliably to freehand sketches, such as those in Fig. 1."

### 2.2    SEMANTIC GENERATION

Several works aim to improve sketch interpretation in generative models by focusing on the semantic content of sketch inputs. We categorize these approaches into two groups: methods that require training and those that are training-free.

#### 2.2.1    TRAINED

**Single-object**    DiffSketching (Wang et al., 2023) targets single-object inputs and relies on a *classifier guidance* and perceptual losses (LPIPS Zhang et al. (2018a) and ResNet He et al. (2016)). This strong reliance on classifier guidance makes it unsuitable for multi-object settings. CLAY (Zhang et al., 2024a) uses dense cross-attention to incorporate sketch conditions into 3D model generation, requiring a substantial number of training parameters. Koley et al. (2024) replace text guidance with a global single-object sketch encoding. To fine-tune the CLIP-based sketch encoder, they align the text and the sketch encoding during training and use a CLIP perceptual loss (Sain et al., 2023). Our work focuses on scene sketches that usually contain multiple objects.

**Multi-object**    Wu et al. (2023) fine-tune a pixel-space diffusion model on FS-COCO freehand sketches (Chowdhury et al., 2022). They train a ResNet-based sketch encoder with additional transformer blocks from scratch, then pass those latent features instead of text to a denoiser. To boost realism on freehand sketches, a discriminator in an image space is trained. Zhang et al. (2024b) propose a multi-step approach that requires fine-tuning a diffusion model for each semi-synthetic scene sketch from SketchyCOCO (Gao et al., 2020), where scenes are composed by assembling individual object sketches. After manually segmenting objects, ControlNet generates their images, and unique identity embeddings are obtained using multi-concept inversion (Avrahami et al., 2023a). During inference, a blended latent diffusion model (Avrahami et al., 2023b) merges object and background representations at high noise levels, while later denoising steps are conditioned on background prompts and object embeddings. However, depending on the noise schedule, outputs may appear unrealistic or fail to reflect the intended object arrangement. Our method also targets freehand scene sketches and is designed for latent diffusion models. It aims at photorealistic generation while closely adhering to sketch guidance, without adding any inference-time overhead.

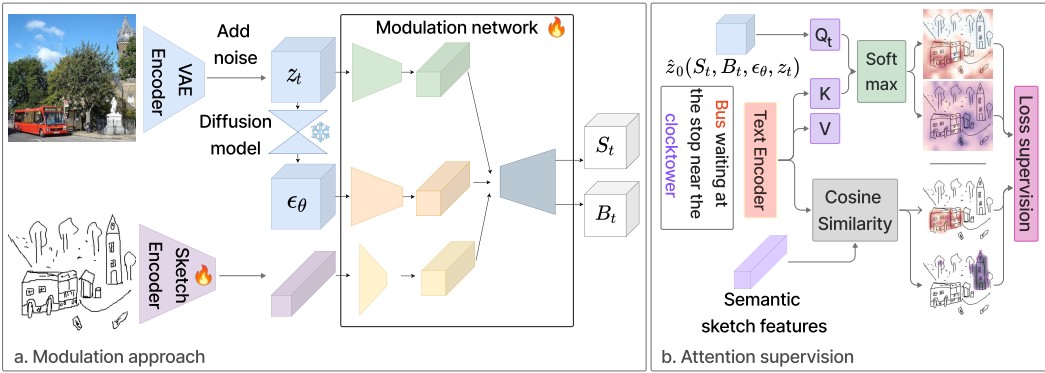

Figure 2: Method overview. **Modulation network (a).** During training, the latent features $z_t$ are generated by applying a noising process to the clean latent $z_0$, which is obtained by encoding the ground-truth image using the VAE encoder. During inference, $z_t$ is instead sampled from a standard Gaussian distribution and then progressively denoised by the diffusion model. Text-conditioned diffusion model generates time-dependent noise $\epsilon_t$, which we modulate relying on semantic sketch features. For details, please refer to Sec. 3.2 and Tab. 4. **Attention supervision (b).** Attention supervision allows us to train on a combination of freehand sketches and sketches algorithmically generated from reference images. It bypasses the need for pixel-aligned ground-truth images — which are unavailable for freehand sketches — and helps the modulation network focus on sketch semantics. For details, please refer to Sec. 3.4.

### 2.2.2 Training-free

Chen (2023) combines ControlNet with *paint-with-words* guidance (Balaji et al., 2022), which uses user-drawn masks to steer cross-attention during inference to add semantic guidance. This approach is sensitive to weighting, requiring careful tuning per category (Fig. 11). Several works rely on the diffusion inversion of a reference sketch (Mo et al., 2024; Ding et al., 2024). FreeControl (Mo et al., 2024) first generates text-conditioned images to extract semantic and appearance bases, then transfers structural guidance by minimizing the distance between the basis coefficients of the generated and reference images. Ding et al. (2024) update noisy latents using KL divergence gradients between cross-attention maps from the current generation and those obtained by diffusion inversion of a reference sketch, showcasing the method only on single-object sketches. We also explore the properties of cross-attention maps, supervising them at training time with the introduced loss.

## 3 Method

In this section, we describe the key components of our method, their motivation, and the necessary background theory. We provide additional implementation details in Secs. 4.1 and 4.2.

### 3.1 Preliminaries on Latent Diffusion Models

Latent Diffusion Models (LDMs) improve the efficiency and scalability of diffusion models by operating in a compressed latent space instead of pixel space. A pre-trained autoencoder first uses its encoder $\mathcal{E}$ to encode input images $x_0 \in \mathbb{R}^{H \times W \times 3}$ into a lower-dimensional latent representation $z_0 = \mathcal{E}(x_0)$.

During training, Gaussian noise is gradually added to a clean latent $z_0$ over discrete timesteps $t \in \{1, \ldots, T\}$, producing a noisy latent $z_t$ at each step. A neural network with parameters $\theta$ is then trained to predict the noise $\epsilon_\theta(z_t, t, c)$ added to $z_0$ at each timestep $t$, optionally conditioned on some context $c$ (e.g., class label or text embedding). The model is trained using the following objective:

$$\mathcal{L}_{\text{noise}} = \mathbb{E}_{z_0, \epsilon, t} \left[ \| \epsilon - \epsilon_\theta(z_t, t, c) \|^2 \right], \tag{1}$$

where $\epsilon \sim \mathcal{N}(0, \mathbf{I})$ is the Gaussian noise added at each timestep according to a predefined noise schedule.

During inference, the model starts from Gaussian noise in the latent space and iteratively denoises it using the learned reverse process. The final output is obtained by decoding the denoised latent $\hat{z}_0(z_t, t)$ using the decoder $\mathcal{D}$: $\hat{x}_0 = \mathcal{D}(\hat{z}_0)$.

## 3.2 MODULATION NETWORK

To enable semantic image generation from abstract freehand scene sketches, we design a modulation network that refines the predicted noise $\epsilon_\theta(z_t, t, c_{text})$ at each timestep, incorporating sketch guidance. Extending the ideas of noise modulation with spatial conditions Ham et al. (2023), we propose a network explicitly designed to exploit semantic sketch features. Our network receives as input the semantic sketch features, the latent $z_t$, and the predicted noise $\epsilon_\theta(\cdot)$ and produces scale $S_t$ and shift $B_t$ maps. These maps are used to modulate the noise prediction as $\epsilon'_\theta(z_t, t, c_{text}, c_{sketch}) = \epsilon_\theta \odot (1 + S_t) + B_t$. The next section Sec. 3.3 details how semantic sketch features are obtained. Here, we focus on the modulation network itself.

Our modulation network is an encoder-decoder CNN architecture. As illustrated in Fig. 2a., each input modality is first processed through a separate downsampling path, projecting it into an embedding space of equal dimensionality. The resulting feature maps are concatenated and passed through a timestep-conditioned convolutional layer, followed by an activation function and another convolutional layer. The fused features are then processed by three upsampling blocks. The final output is split into scale $S_t \in \mathbb{R}^{H \times W \times 4}$ and shift $B_t \in \mathbb{R}^{H \times W \times 4}$ latent maps.

The model is trained with the loss defined in Eq. (1) and additional regularization losses. Similarly to Ham et al. (2023), we apply $\mathcal{L}_1$ regularization on shift and scale parameters:

$$\mathcal{L}_1^{scale} = \|S_t\|_1, \quad \mathcal{L}_1^{shift} = \|B_t\|_1. \tag{2}$$

To promote more expressive and diverse transformations, we encourage variability in the predicted scale and shift maps by penalizing low variance across their values:

$$\mathcal{L}_{var} = -(\sigma(S_t) + \sigma(B_t)), \tag{3}$$

where $\sigma$ is the standard deviation computed over the elements of the modulation maps $S_t$ and $B_t$.

## 3.3 SEMANTIC SKETCH FEATURES

To extract semantic features from freehand sketches, we leverage a pre-trained CLIP-based encoder (Bourouis et al., 2024). This encoder was trained on freehand sketches from the FS-COCO dataset (Chowdhury et al., 2022) to perform open-vocabulary semantic segmentation of scene sketches. However, for the task of sketch-conditioned image generation, these features may lack sufficient fine-grained detail. Empirically, we found that fine-tuning the last few layers (three in our case) of the encoder significantly improves the alignment between the generated image and the input sketch.

## 3.4 TRAINING WITH FREEHAND SKETCHES

Training diffusion models with freehand sketches presents a key challenge: pixel-aligned, realistic reference images do not exist, and as discussed in Sec. 1, such images can rarely exist in practice. Using misaligned reference images, such as those in the FS-COCO dataset (Fig. 1), introduces ambiguity in the standard denoising objective (Eq. (1)). To overcome this, we propose a loss function that bypasses pixel-level correspondence and instead focuses on preserving the sketch's semantic structure.

Before introducing our loss, we briefly review how text conditioning is commonly implemented in generative models via a cross-attention mechanism:

$$\text{Attention}(Q, K, V) = \text{Softmax}\left(\frac{QK^\top}{\sqrt{d}}\right)V = MV, \tag{4}$$

where the query matrix $Q$ is derived from the latent representation $z$, while the key $K$ and value $V$ matrices are obtained from the text embedding $c$. The latent dimensionality is denoted by $d$. Each entry $M_{ij}$ in the attention matrix $M$ captures the influence of the $j$-th text token on the $i$-th image patch, therefore defining in which spatial locations each particular semantic concept appears.

We leverage the observation that the semantic encoder used in our modulation network, originally trained for segmentation, provides a strong spatial signal indicating the location and identity of objects in the sketch (see Fig. 2b.). Therefore, we use the features of the original pre-trained sketch encoder (Bourouis et al., 2024) to compute 'ground-truth' attention maps $M_{\mathrm{grth}}$. For synthetic sketches, we rely on the available segmentation maps in the MS-COCO dataset (Lin et al., 2014).

From the modulated noise $\epsilon'_\theta(z_t, t, c_{text}, c_{sketch})$, we compute the denoised latent $\hat{z}_0(z_t, t, \epsilon'_\theta(\cdot))$. We then pass $\hat{z}_0(\cdot)$ through the denoising network (with text conditioning) to extract cross-attention maps $M$ at multiple layers. We supervise these cross-attention maps $M$ using ground-truth attention maps $M_{\mathrm{grth}}$, following the loss formulation introduced by Sun et al. (2024) in the context of layout-based image generation:

$$\mathcal{L}_{\mathrm{attn}} = \sum_{\gamma \in \Gamma} \sum_{i \in \mathcal{I}} \sum_{b_i \in \mathcal{B}} \left( 1 - \left( \frac{\sum_{p \sim b_i} M_{pi}^{(\gamma)}}{\sum_p M_{pi}^{(\gamma)}} \right)^2 - \lambda_{\mathrm{reg}} \sum_{p \sim b_i} M_{pi}^{(\gamma)} \right), \tag{5}$$

here, $M_{pi}^{(\gamma)}$ denotes the attention value at pixel $p$ for object $P_i$ in layer $\gamma$. The set $\mathcal{B}$ contains spatial regions derived from $M_{\mathrm{grth}}$, and $\mathcal{I}$ is the set of valid text token indices. The loss encourages alignment of attention with the target layout while the regularization term, weighted by $\lambda_{\mathrm{reg}}$, prevents attention from leaking outside the target regions. The summation over $\gamma \in \Gamma$ aggregates contributions across multiple attention layers.

We define our total objective as a weighted sum of four loss components:

$$\mathcal{L}_{\mathrm{total}} = \lambda_0 \mathcal{L}_{noise} + \lambda_1 L_{\mathrm{attn}} + \lambda_2 \left( \mathcal{L}_1^{\mathrm{scale}} + \mathcal{L}_1^{\mathrm{shift}} + \mathcal{L}_{\mathrm{var}} \right). \tag{6}$$

## 4 Experiments

### 4.1 Training and Test Data

For training, we use the FS-COCO dataset (Chowdhury et al., 2022), as the only available dataset of freehand sketches. FS-COCO augments a subset of 10,000 MS-COCO (Lin et al., 2014) images with paired freehand sketches and captions that describe the sketches. We use the split of FS-COCO from (Bourouis et al., 2024), as it guarantees that the sketches in the test set contain a subset of sketch styles not seen during training. In total, we use 9,525 samples for training and 475 for testing. For each image in the FS-COCO training set, we additionally generate a synthetic sketch using the method by Su et al. (2023). We conduct all analyses in this section based on the methods' performance on the 475 freehand sketches.

### 4.2 Implementation details

We obtain $M_{\mathrm{grth}}$ for freehand sketches by computing the similarity between each feature patch, extracted using a pretrained semantic sketch encoder (Bourouis et al., 2024), and the CLIP text embedding of object $P_i$. A threshold of 0.5 is then applied to produce a binary mask. We set $\lambda_0, \lambda_1 = 1.0$ and $\lambda_2 = 0.1$. Our batch consists of 50% freehand and 50% synthetic sketches. For freehand sketches, we set $\lambda_0 = 0.0$ and use captions from the FS-COCO dataset. Training is restricted to the top 10% of diffusion timesteps, corresponding to high-noise regimes which are known to control the overall semantic structure. During inference, only those 10% noise steps are modulated. We ablate this choice in Sec. D.3. As the diffusion backbone for our method in this section, we use Stable Diffusion 2.1 (SD2.1) (Rombach et al., 2022), chosen for its relatively lightweight architecture and still strong generative performance. During training, the total number of noise steps is $T = 1000$, and during inference, it is $T = 50$. We train on a single A100 GPU. We provide additional results with the SDXL backbone in Sec. A.3.

### 4.3 Evaluation Metrics

We evaluate our method using three metrics: (i) FID (Heusel et al., 2017), to assess the visual quality and diversity of generated images against real ones; (ii) sketch-image similarity, computed via cosine similarity between sketch encoder and CLIP image embeddings, assessing consistency with the input sketch; and (iii) LPIPS (Zhang et al., 2018b), to measure perceptual similarity between generated and reference images, capturing human-aligned visual fidelity beyond pixel-level differences..

### 4.4 COMPARISON TO STATE OF THE ART

In this section, we focus our evaluation on performance on freehand sketches; additional comparisons on synthetic sketches are provided in Sec. D.2. Results of the user study where participants rank outputs from different methods are provided in Sec. A.2.

#### 4.4.1 BASELINES AND CHOICE OF THEIR OPTIMAL PARAMETER SETTINGS

We compare our approach with several widely adopted training-based conditional diffusion methods: ControlNet (Zhang et al., 2023) with two diffusion model backbones SD2.1 (Rombach et al., 2022) and SDXL (Podell et al., 2023), ControlNext (Peng et al., 2024), T2I-Adapter (Mou et al., 2023), and SG (Voynov et al., 2023). Note that these methods were originally trained on sketch styles that match our synthetic sketches. We refer to this setting as *zero-shot*. We also compare our method with the inference-time method FreeControl (Mo et al., 2024). Additionally, in Sec. A.5 we compare with FluxKontext (bfl).

ControlNet and T2I-Adapter include control parameters that balance sketch fidelity and image realism. ControlNet supports a scale factor $w$ for the conditional branch connections. For T2I-Adapter, there are two parameters Jiang et al. (2025): $s$, which controls how much the conditional features are added in residual connections, and $\tau$, which controls for which time steps $t > T(1 - \tau)$ the spatial guidance is added, where $T$ is the total number of steps used.

Since our goal is to generate images with the best trade off between photorealism and alignment with the input sketch, we conduct a user study to identify the optimal parameter setting for each method on our freehand sketches. When multiple settings yield similar user rankings, we additionally consider quantitative metrics to select the configuration that provides the best overall performance. We found the following setting to be optimal: $w = 0.4$ for ControlNet SD2.1, $w = 0.6$ for ControlNet SDXL, and two sets of parameters for T2I $s = 0.8, \tau = 0.2$ and $s = 0.8, \tau = 0.4$. We refer the reader to Sec. C.1 for more details, and visual comparisons among different parameter settings.

We begin by evaluating all models in a zero-shot setting, using pre-trained weights without modification. To enable fair comparison, we further fine-tune each baseline on our training set using two configurations: (i) optimization with the denoising loss alone (Eq. (1)), and (ii) joint optimization with both the denoising and attention supervision losses (Eq. (5)).

#### 4.4.2 QUANTITATIVE COMPARISON

Table 1 shows that our method achieves the best FID, CLIP similarity, and LPIPS sketch-to-image alignment among all baselines. This demonstrates our model's stronger ability to retain semantic and structural cues from abstract sketches.

For ControlNet, we first perform naïve fine-tuning independently on synthetic and freehand sketches from our dataset, treating them as separate training sets. On synthetic sketches, while this leads to a slight increase in FID, both CLIP similarity and LPIPS improve, suggesting better adherence to the sketch input, expected from fine-tuning. The higher FID may stem from the image quality within the MS-COCO dataset, which we use for both training and fine-tuning. However, when fine-tuning on freehand sketches, all metrics degrade. We attribute this to the lack of pixel-level alignment in freehand sketches, which limits model's ability to learn effective spatial guidance from them. When we add our proposed attention loss and train on a mix of synthetic and freehand sketches, FID decreases only slightly, while both CLIP similarity and LPIPS improve substantially.

For T2I, $s = 0.8, \tau = 0.2$ results in better FID scores, close to those of ControlNet, but a poor alignment with freehand sketches. Setting $s = 0.8, \tau = 0.4$ characterizes in lower FID scores, but better alignment with input sketches. In both cases, naïve fine-tuning on a mix of freehand and synthetic sketches improves the performance according to all metrics. Adding the attention loss further improves performance, with a particularly pronounced effect on FID scores. These results show the importance of the proposed attention loss when targeting freehand sketch inputs.

ControlNet SDXL (zero-shot) results in images of poor quality, poorly aligned with input sketches. Finally, ControlNext Peng et al. (2024) (Fig. 16) generates visually appealing images but often fails to accurately follow the structural guidance provided by the sketch input.

| Method | Setup | FID↓ | CLIP↑ | LPIPS↓ |
|---|---|---|---|---|
| CntrlNet SD2.1 (Zhang et al., 2023) | Zero-shot | 135.595 | 1.136 | 0.773 |
|  | $\mathcal{L}_{\text{noise}}$ only (syn) | 136.821 | 1.141 | 0.771 |
|  | $\mathcal{L}_{\text{noise}}$ only (free) | 139.872 | 1.042 | 0.789 |
|  | $\mathcal{L}_{\text{noise}} + \mathcal{L}_{\text{attn}}$ | 135.891 | 1.196 | 0.768 |
| T2I Adapter $s = 0.8\,\tau = 0.2$ (Mou et al., 2023) | Zero-shot | 144.329 | -0.203 | 0.813 |
|  | $\mathcal{L}_{\text{noise}}$ only | 138.479 | 0.318 | 0.779 |
|  | $\mathcal{L}_{\text{noise}} + \mathcal{L}_{\text{attn}}$ | 137.982 | 0.391 | 0.774 |
| T2I Adapter $s = 0.8\,\tau = 0.4$ (Mou et al., 2023) | Zero-shot | 159.816 | 0.213 | 0.819 |
|  | $\mathcal{L}_{\text{noise}}$ only | 151.736 | 0.426 | 0.781 |
|  | $\mathcal{L}_{\text{noise}} + \mathcal{L}_{\text{attn}}$ | 139.568 | 0.454 | 0.778 |
| CntrlNet SDXL (Zhang et al., 2023) | Zero-shot | 174.462 | 0.027 | 0.825 |
| CntrlNext SDXL (Peng et al., 2024) | Zero-shot | 134.094 | 0.909 | 0.774 |
| SG (Voynov et al., 2023) | Zero-shot | 137.381 | 1.043 | 0.782 |
| FreeControl (Mo et al., 2024) | Inference-time | 141.632 | 1.089 | 0.793 |
| **Ours** | Full | **121.973** | **1.291** | **0.739** |

Table 1: Comparison on 475 test sketches from the FS-COCO dataset (Chowdhury et al., 2022) with baseline methods with different fine-tuning setups. The used metrics are summarized in Sec. 4.3.

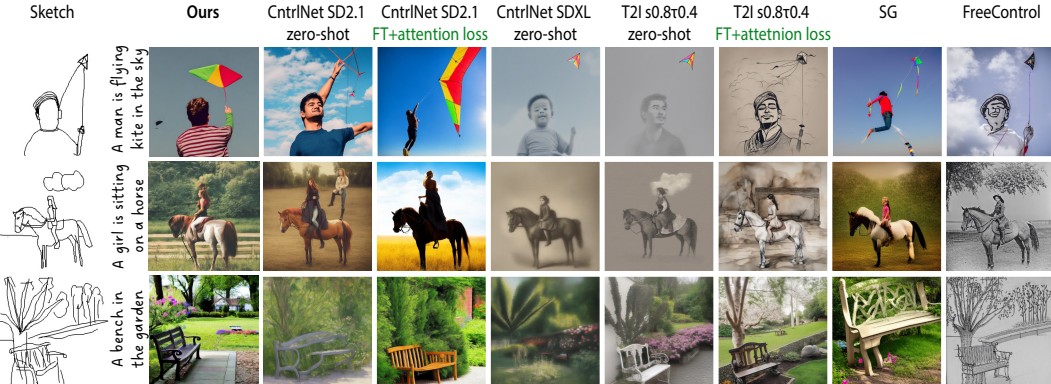

Figure 3: Visual comparison between our method and baselines in a zero-shot setting (using weights of pretrained models) and with the ones fine-tuned with the proposed attention loss on a mix of freehand and synthetic sketches.

### 4.4.3 QUALITATIVE COMPARISON

As shown in Figs. 3, 6, and 7, our method produces photorealistic images that faithfully capture the key structural elements of the input sketches. ControlNet SD2.1 frequently fails in the zero-shot setting on freehand sketches, often generating implausible images that do not adhere to the input guidance. While ControlNet SDXL might better adhere to the sketch input than its SD2.1 counterpart, its outputs tend to be desaturated, blurry, or exhibit a cartoon-like appearance. Lower values of the weighting factor $w$ would yield higher visual quality, but at the cost of reduced sketch adherence, as illustrated in Fig. 14. Similarly, T2I variants with the user-study-based parameter settings overall follow sketch inputs, but produce washed-out, often cartoonish images. Higher values of $\tau$ lead to better sketch adherence, but lower image quality.

Fine-tuning on a mix of synthetic and freehand sketches using our attention loss leads to improved visual adherence to the input sketches and more vivid color palettes compared to zero-shot counterparts. However, the results remain less consistent and exhibit a higher prevalence of artifacts compared to our full method.

### 4.5 ABLATIONS

#### 4.5.1 ROLE OF SKETCH REPRESENTATION

First, we ablate the choice of sketch representation used in our modulation network. A common baseline is to encode sketches using the VAE encoder shared with image inputs, as done in ControlNet (Zhang et al., 2023). In this setup, we introduce a separate branch for VAE-based sketch

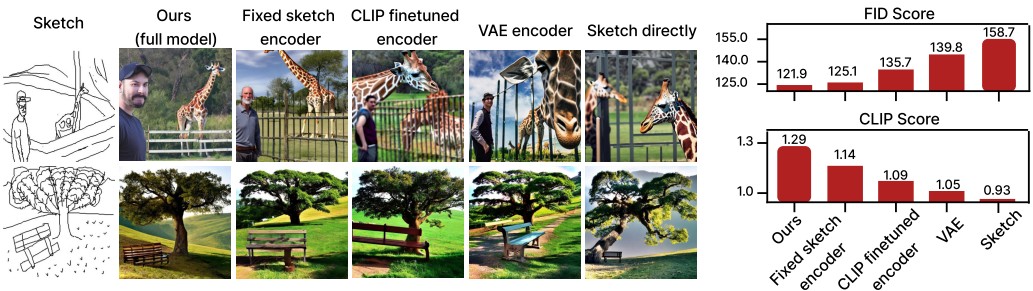

Figure 4: Quantitative evaluation of the role of sketch representation. Please refer to Sec. 4.5.1 for the details.

features. This branch mirrors the architecture of the noise and latent branches but employs its own set of weights. As shown in Fig. 4, this leads to significantly higher FID and lower CLIP similarity scores, indicating degraded performance. In this configuration, the modulation network underperforms compared to ControlNet (Tab. 1).

Next, we experiment with directly using sketches as input to the modulation network, following the approach of Ham et al. (2023). Unlike our method, their modulation network employs a single-branch UNet (Ronneberger et al., 2015), where all inputs are concatenated before being processed. In contrast, we find that using separate downsampling branches for each modality leads to improved performance. In this experiment, sketches are encoded using a dedicated branch consisting of progressive downsampling and channel expansion via convolutional layers with SiLU activations, followed by several time-conditioned convolutional blocks. As shown in Fig. 4, this setup results in the worst performance, suggesting that it fails to fully leverage the information encoded in sketches.

Finally, Fig. 4 underscores the importance of our sketch encoder fine-tuning strategy. Even without fine-tuning, our model outperforms all baselines (Tab. 1). However, the original CLIP encoder (Radford et al., 2021), even with fine-tuning, fails to match our approach.

### 4.5.2 ROLE OF EACH OF THE LOSSES

We first ablate the role of attention loss by training a variant of our model with $\lambda_0 = 1$ and $\lambda_2 = 0$ for both synthetic and free-hand sketches, effectively reducing the objective to a standard diffusion loss with regularization terms without attention supervision. As shown in Fig. 5, removing the attention loss leads to a modest improvement in FID score, but significantly reduces sketch-image alignment.

Removing $\mathcal{L}_{\mathrm{var}}$ degrades both image quality, as indicated by higher FID scores, and sketch-image alignment, as reflected in lower CLIP similarity (Fig. 5). Thus, this loss results in higher expressivity of the model, facilitating the learning process.

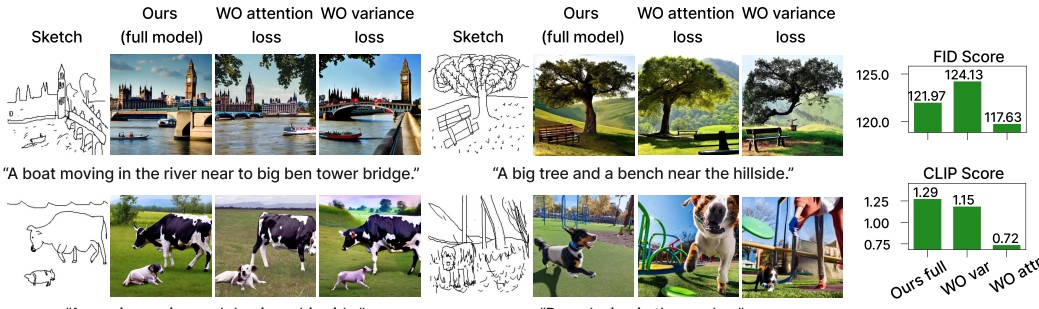

Figure 5: Qualitative and quantitative evaluation when we (i) remove the attention loss $\mathcal{L}_{\mathrm{attn}}$ (Eq. (5)), 'WO attention loss', and (ii) when we remove $\mathcal{L}_{\mathrm{var}}$ (Eq. (3)), 'WO variance loss'. Please refer to Sec. 4.5.2 for the detailed discussion. Captions are taken as-is from the FSCOCO dataset.

## 5 CONCLUSIONS

Our main contribution lies in shifting the focus of pixel-aligned spatial conditioning to semantic-aware generation, which emphasizes what is in the scene and where it is located. We address the under-explored problem of generating images from real freehand scene sketches, which often exhibit abstraction and distortion. We show how to extract semantic features using an appropriate encoder and condition a diffusion model on them without altering the backbone architecture. Our design, including our proposed attention loss, allows us to effectively train on freehand sketches, consistently improving performance across baseline conditioning methods.

### ACKNOWLEDGMENTS

We acknowledge the support of the Natural Sciences and Engineering Research Council of Canada (NSERC) under Grant No.: RGPIN-2024-04968 ("Modelling and animation via intuitive input"), the NSERC — Fonds de recherche du Québec — Nature et technologies (FRQNT) NOVA Grant No. 314090, FRQNT team grant No. 361570, and a gift from Adobe.

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

# A ADDITIONAL EVALUATIONS

## A.1 ADDITIONAL VISUAL RESULTS

Additional visual results, comparing our method and baselines, are provided in Figs. 6 and 7.

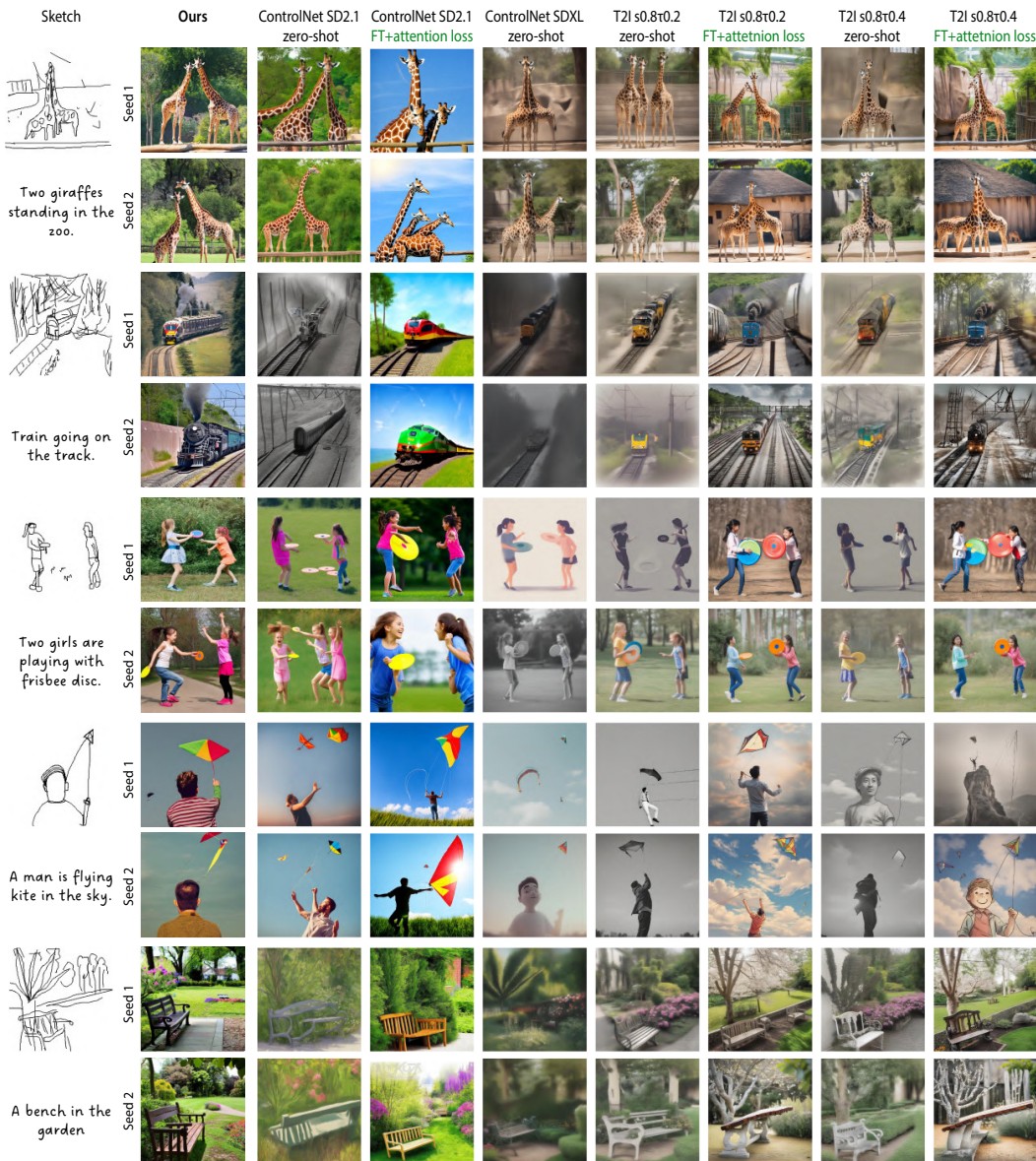

Figure 6: Qualitative results comparison between our method and baselines. Please see Sec. 4.4.2 for the detailed discussion. Sketch and text are both passed in as inputs.

## A.2 USER PREFERENCE STUDY

We conducted a user study involving 23 participants to subjectively evaluate generation quality. We compared our method against the best-performing baselines: ControlNet SD2.1, ControlNet SDXL, and T2I-Adapter. The sketches were randomly selected from the entire test set for each participant. Each participant was asked to evaluate 20 pairwise comparisons: ours against one of the baselines. The same sketch was not shown twice to users. We asked participants to pick an image according to each of the following criteria: (i) *Which of the two images looks more photorealistic (i.e., most like*

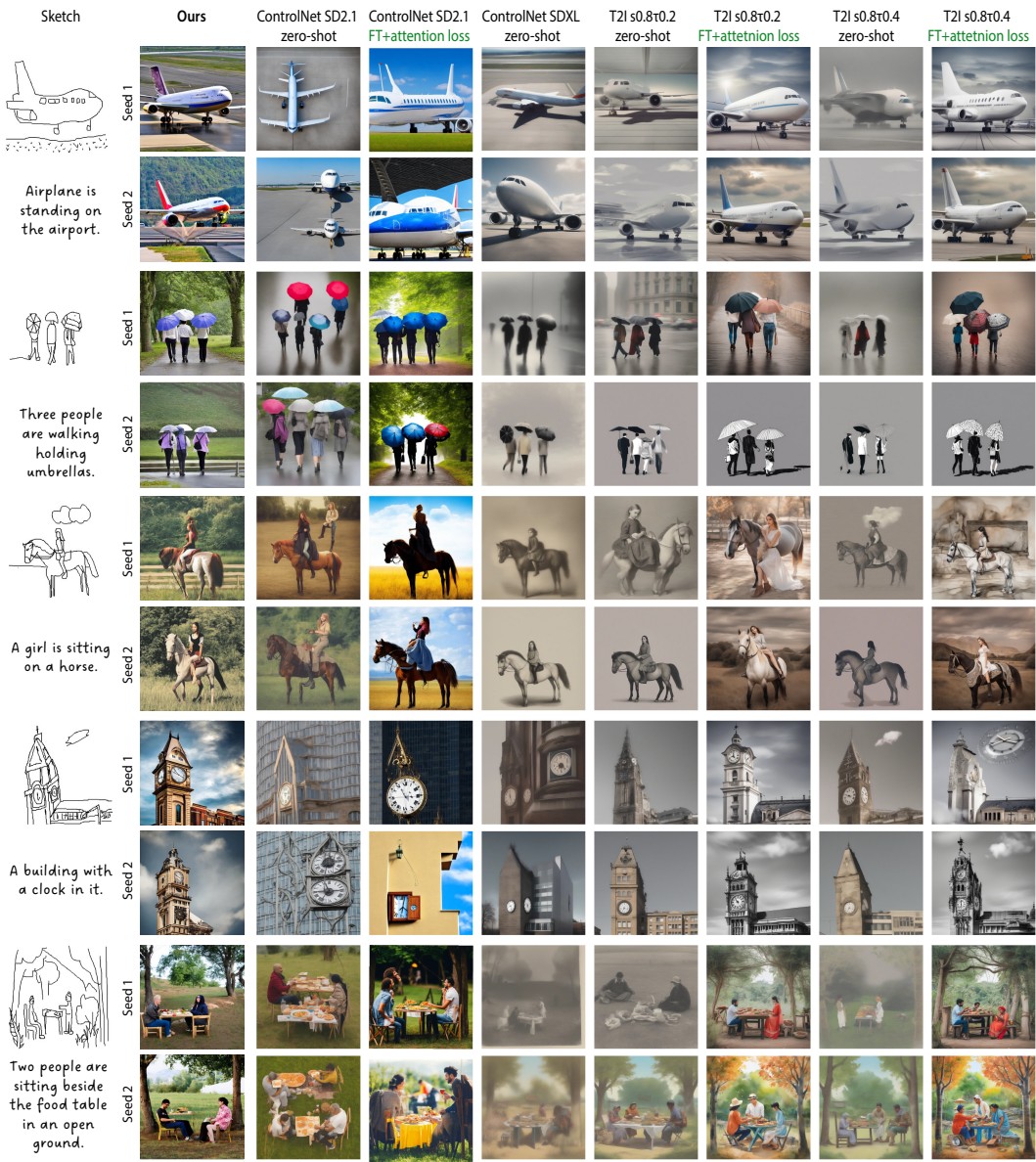

Figure 7: Qualitative results comparison between our method and baselines. Please see Sec. 4.4.2 for the detailed discussion. Sketch and text are both passed in as inputs.

*a real photograph)?* (ii) *Which image is more similar to the sketch in terms of layout and overall structure?* (ii) *Which image is the better trade-off between photorealism and similarity to the sketch?*

| **Baseline** | **Photorealism (%)** | | | **Sketch similarity (%)** | | | **Best trade-off (%)** | | |
|---|---|---|---|---|---|---|---|---|---|
| | **Ours** | **Base** | **Und.** | **Ours** | **Base** | **Und.** | **Ours** | **Base** | **Und.** |
| SDXL T2I s=0.8 $\tau$=0.2 | **47.2** | 34.9 | 17.9 | **46.4** | 37.1 | 16.5 | **66.0** | 20.6 | 13.4 |
| SDXL T2I s=0.8 $\tau$=0.4 | **47.4** | 39.2 | 13.4 | **55.0** | 35.8 | 9.2 | **71.0** | 22.0 | 7.0 |
| ControlNet SDXL | 40.4 | **45.2** | 14.4 | **54.0** | 37.0 | 9.0 | **54.4** | 32.2 | 13.3 |
| ControlNet SD2 | **70.0** | 21.2 | 8.8 | **64.8** | 30.8 | 4.4 | **88.8** | 10.0 | 1.2 |

Table 2: Pairwise user study preferences. Please see Sec. A.2 for the details.

As shown in Tab. 2, our method is consistently preferred across all baselines for the third question ("best trade-off"). While it trails slightly behind ControlNet SDXL in perceived photorealism –

| | Method | Setup | FID↓ | CLIP↑ | LPIPS↓ |
|---|---|---|---|---|---|
| #1 | CntrlNet SDXL (Zhang et al., 2023) | Zero-shot | 174.462 | 0.027 | 0.825 |
| #2 | | ControlNet branch LoRA | 177.243 | 0.046 | 0.813 |
| #3 | | ControlNet branch LoRA + main branch LoRA | 189.566 | 0.043 | 0.779 |
| #4 | **Ours SD2.1** | Modulation network training | 121.973 | 1.291 | 0.739 |
| #5 | **Ours SDXL** | Modulation network training | **117.025** | **1.331** | **0.708** |
| #6 | **Ours SDXL** | Modulation network training + backbone LoRA fine-tuning | 138.213 | 1.162 | 0.759 |

Table 3: Comparison on 475 test sketches from the FS-COCO dataset (Chowdhury et al., 2022), averaged over three seeds. *(#4) Ours SD2.1* is the main model used throughout the paper, where we train only the modulation network and the last three layers of the sketch encoder with all the proposed losses as described in the main paper. All remaining variants use the SDXL backbone. *(#5) Ours SDXL (Modulation network training)* differs from *(#4) Ours SD2.1 (Modulation network training)* only in the backbone, while the training setup remains the same. *(#6) Ours SDXL (Modulation network training+backbone LoRA fine-tuning)* adds additional LoRA fine-tuning of the backbone SDXL model.

For ControlNet SDXL, LoRA fine-tuning is applied either to the ControlNet branch alone (#2) or to both the ControlNet and main diffusion branches (#3). The used metrics are summarized in Sec. 4.3. Analysis is provided in Sec. A.3.

likely due to the latter's stronger diffusion backbone – it is strongly favored over ControlNet SD2.1, which shares the same backbone. Our method is also consistently rated as producing images that better match the input sketches.

## A.3 TRAINING WITH SDXL BACKBONE

In the main paper, we use the SD2.1 backbone with our method due to its lower computational requirements and faster iteration cycles. To demonstrate that our method is backbone agnostic, we additionally train our pipeline with the SDXL backbone (Tab. 3, Fig. 8). Compared to SD2.1, the SDXL backbone employs a substantially larger and more expressive architecture, offering improved generative fidelity but requiring significantly more computational resources.

We previously showed that our method with the SD2.1 backbone outperforms all alternatives. Here we show that using a stronger backbone further improves our method across all three metrics: FID, CLIP, and LPIPS (Tab. 3: #5 vs. #4). Switching from SD2.1 to SDXL reduces FID from 121.973 to **117.025**, increases CLIP sketch–image similarity from 1.291 to **1.331**, and decreases LPIPS from 0.739 to **0.708**. These results demonstrate that our noise-modulation strategy, together with attention supervision, generalizes effectively to larger backbones without requiring any fine-tuning of the backbone itself. Consequently, our method serves as an efficient, lightweight conditioning mechanism.

We additionally experiment with applying LoRA fine-tuning to the SDXL backbone alongside training our modulation network, which serves as a conditioning mechanism. This approach results in worse performance than training the modulation network alone (compare the last two rows in Tab. 3). We hypothesize that this is due to slower training dynamics as the number of trainable parameters increases. All baselines in this table are trained for the same number of iterations (5K).

## A.4 CONTROLNET WITH LORA FINE-TUNING

As our method introduces a lightweight conditioning mechanism that optimizes only a small number of parameters (9.4M), we compare it against ControlNet under a LoRA fine-tuning setup. Note that we begin with the ControlNet variant whose control branch has already been fine-tuned on synthetic sketches generated programmatically from reference images. Fine-tuning the full ControlNet branch requires training 1.5B parameters. In this setting, we only use the original diffusion loss, but train on a combination of freehand and synthetic sketches.

As shown in Tab. 3, LoRA performs poorly in this setting. Applying LoRA only to the ControlNet branch slightly improves CLIP similarity (0.027 → 0.046) but results in worse FID (174.462 →

177.243). Extending LoRA to both the ControlNet and the diffusion backbone further degrades FID to 189.566, while CLIP similarity remains low (0.043). Applying LoRA only to the ControlNet branch requires training 12M parameters. These observations are consistent with the experiments reported in the main paper, where fine-tuning all weights of the ControlNet (SD2.1) branch on a combination of freehand and synthetic sketches proved suboptimal. These experiments show that naive fine-tuning strategies with freehand sketches degrade the visual quality of the generated images.

Both Tab. 3 and Fig. 8 demonstrate that our conditional mechanism, implemented via the proposed modulation network and combined with the attention supervision loss, achieves superior results regardless of the backbone. Furthermore, using a more powerful backbone leads to improved performance. *Ours (SDXL)*, similarly to *Ours (SD2.1)*, provides a favorable tradeoff between sketch adherence and visual realism.

**Implementation details**   In all LoRA experiments, we use a rank of 16, an alpha of 32, and a dropout of 0.1, with a learning rate of 0.0001. Training is performed for 5K iterations with a batch size of 128 on 4 A100 GPUs. Both training and evaluation are conducted on the same dataset as in the main paper, consisting of a mixture of freehand and synthetic sketches (see Sec. 4.1).

## A.5    Comparison with Flux.1 Kontext [dev]

Additionally, we compare our method with the recent universal image editing model FluxKontext (bfl), based on FLUX. Although this model is not specifically trained for sketch-to-image conversion, it benefits from training on a massive dataset. We prompted the FluxKontext model with *"Convert a sketch to a photorealistic image of "* concatenated with the same prompt as we pass to our model. We experimented with a set of prompts for the FluxKontext model and empirically found that this one yielded the best results.

As shown in Fig. 9, our method demonstrates clear advantages over the state-of-the-art universal image editing model Flux.1 Kontext [dev]. Our model produces more photorealistic output, better follows input sketches, and remains robust even with distorted or cluttered sketches. In the bear and plane examples on the left in Fig. 9, we can see that, unlike Flux.1 Kontext [dev], our method follows the sketch well and produces variations in visual details and colors across different seeds. In the dog example, Flux Kontext fails to localize the dog in the grass.

Quantitatively, we achieve a lower FID (121.973 vs. 132.487) and a higher CLIP sketch–image similarity score (1.291 vs. 1.226), validating both the superior realism and the stronger sketch alignment for our method.

## A.6    Generalization to QuickDraw dataset

To assess generation beyond scene-level sketches, we also evaluate our method on single-object sketches from the QuickDraw dataset Jongejan et al. (2017). To illustrate how our method performs on single-object sketches relative to specialized state-of-the-art approaches, we include a visual comparison with results from AboutYourSketch [3], a recent method specifically designed for single-object sketches and built on the SD1.5 backbone. As shown in Fig. 10, our approach achieves visual quality comparable to the specialized single-object method *AboutYourSketch* Koley et al. (2024), despite never being trained on isolated-object data. Our model preserves object shape and produces photorealistic textures on par with the dedicated baseline, while maintaining the broader flexibility needed for complex multi-object scenes. For our method, we use the prompt: *"A photo of a `<category>`"*. For *AboutYourSketch*, the visual results are taken directly from the paper. A numerical comparison was not possible due to the unavailability of the baseline's code.

## A.7    Combining ControlNet with paint-with-words guidance (Balaji et al., 2022).

A public implementation (Chen, 2023) combines ControlNet with *paint-with-words* guidance (Balaji et al., 2022), which uses user-drawn masks to steer cross-attention during inference to add semantic guidance. Fig. 11 shows that the performance is dependent on the setting of control parameters

for each object category, while our method produces consistent results, without the need for the segmentation masks and object-specific parameter tuning.

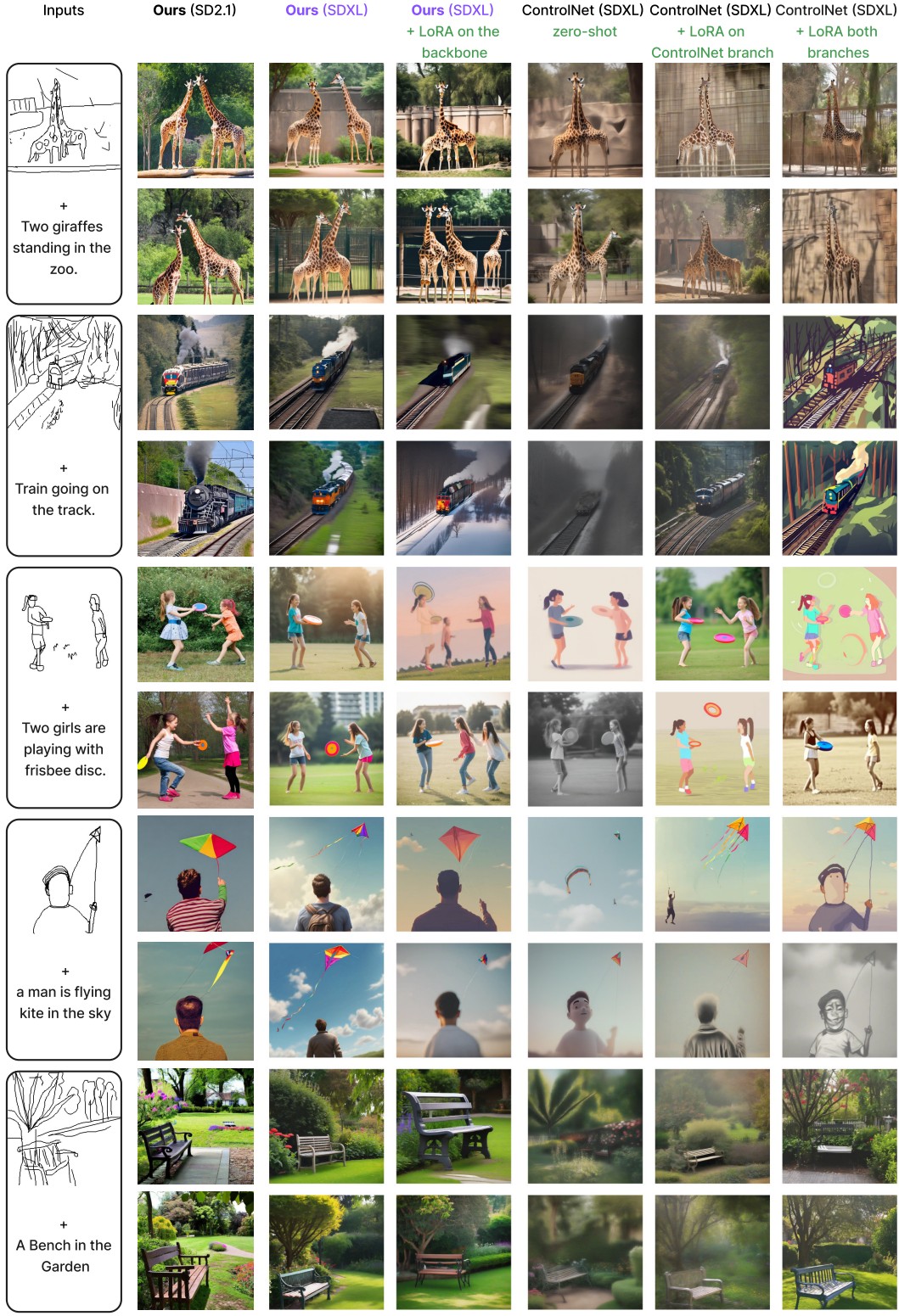

Figure 8: Qualitative comparison of our method using the SD2.1 backbone, the SDXL backbone, and the SDXL backbone with LoRA fine-tuning of the backbone. We also compare against Control-Net (SDXL backbone) in three configurations: zero-shot, LoRA applied to the control branch, and LoRA applied to both the diffusion and control branches. All methods are conditioned on the input sketches and textual captions shown in the figure. **The results are shown for two different seeds.**

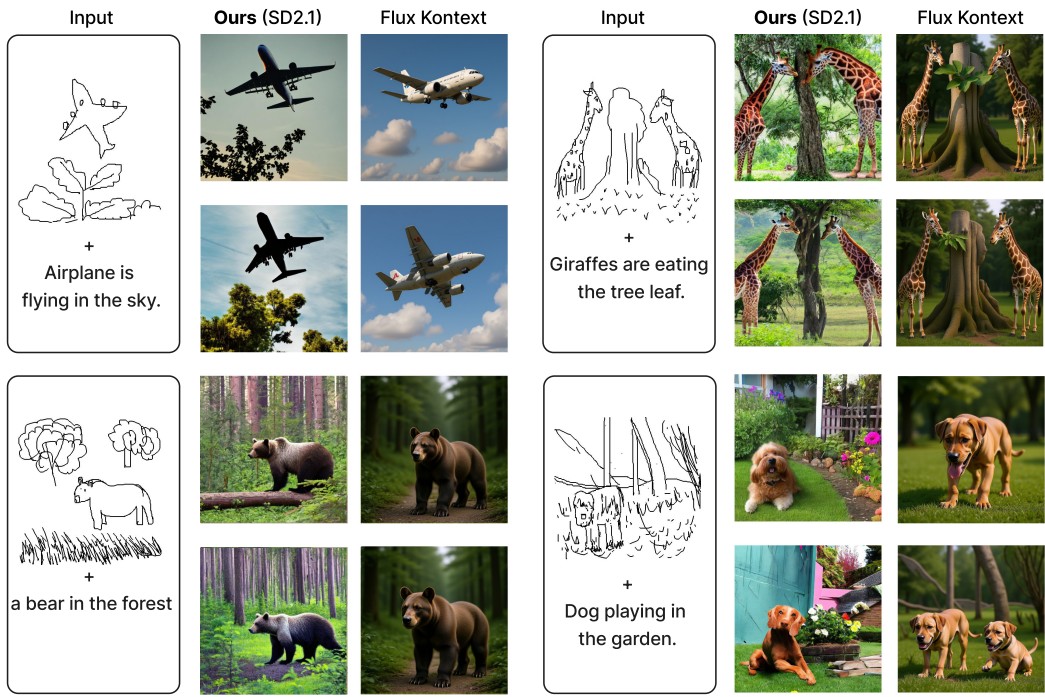

Figure 9: Qualitative results comparison between our method and Flux.1 Kontext [dev]. Our method is conditioned on sketches and textual captions, shown in the figure. We prompted the FluxKontext model with *"Convert a sketch to a photorealistic image of ..."*, where instead of *"..."* we use the same prompt as we pass to our model. **The results are shown for two different seeds.**

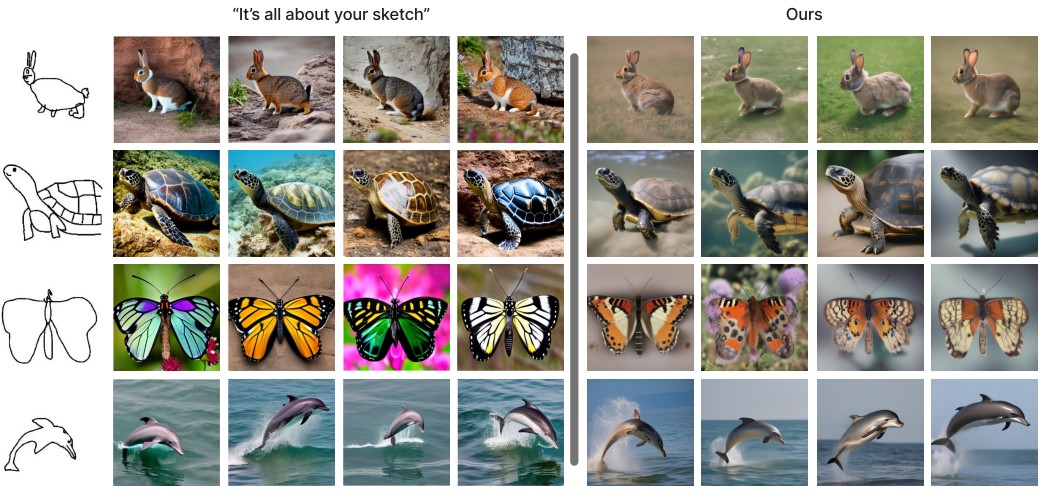

Figure 10: A visual comparison between our method and a specialized single object sketch-based image generation method Koley et al. (2024) on the sketches from the QuickDraw dataset Jongejan et al. (2017).

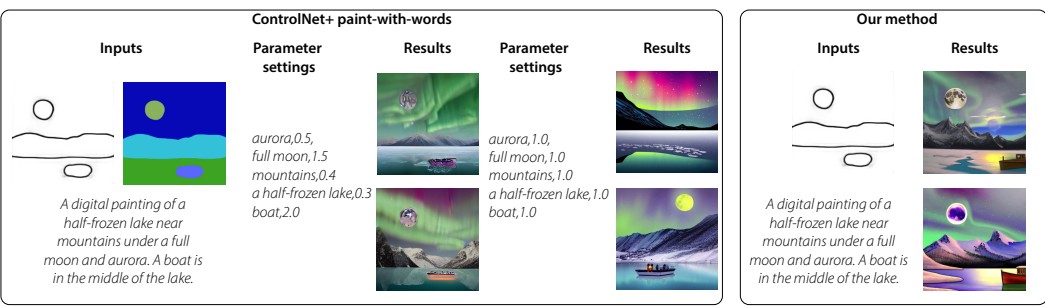

Figure 11: A visual comparison between our method and a public implementation Chen (2023) of the method that combines ControlNet with *paint-with-words* guidance Balaji et al. (2022).

# B IMPLEMENTATION DETAILS

## B.1 MODULATION NETWORK

The detailed design of our modulation network is shown in Tab. 4. Our sketch encodings have dimensions $512 \times 14 \times 14$, where $512$ denotes the number of channels. The VAE latent representations are of size $4 \times 128 \times 128$, where $4$ is the number of channels.

| Path/Block | Layer/Operation | Input Ch. | Output Ch. | Spatial Change |
|---|---|---|---|---|
| | DoubleConv | 512 | 256 | - |
| Sketch Path | DoubleConv | 256 | 128 | - |
| | Interpolate (bilinear) | 128 | 128 | Match latent/noise size |
| | DoubleConv | 4 | 16 | - |
| | MaxPool2d (2) | 16 | 16 | /2 |
| | DoubleConv | 16 | 32 | - |
| Noise $\epsilon$ Path | MaxPool2d (2) | 32 | 32 | /2 |
| | DoubleConv | 32 | 64 | - |
| | MaxPool2d (2) | 64 | 64 | /2 |
| | DoubleConv | 64 | 128 | - |
| | DoubleConv | 4 | 16 | - |
| | MaxPool2d (2) | 16 | 16 | /2 |
| | DoubleConv | 16 | 32 | - |
| Latent $z_t$ Path | MaxPool2d (2) | 32 | 32 | /2 |
| | DoubleConv | 32 | 64 | - |
| | MaxPool2d (2) | 64 | 64 | /2 |
| | DoubleConv | 64 | 128 | - |
| Fusion | Concat [s2, p4, l4] | 384 | - | - |
| | DoubleConv (fusion) | 384 | 256 | - |
| | ConvTranspose2d (up1) | 256 | 128 | ×2 |
| | DoubleConv (up_conv1) | 128 | 64 | - |
| Upsampling | ConvTranspose2d (up2) | 64 | 32 | ×2 |
| | DoubleConv (up_conv2) | 32 | 16 | - |
| | ConvTranspose2d (up3) | 16 | 8 | ×2 |
| | DoubleConv (up_conv3) | 8 | 8 | - |
| Final | Conv2d (final) | 8 | 8 | - |
| | torch.chunk | 8 | 4+4 | - |

Table 4: Architecture of the proposed modulation network.

## B.2 CONTROLNET AND T2I ADAPTER FINE-TUNING

We start from pretrained *sd21-controlnet-scribble* Zhang (2023) and finetune it on our dataset Sec. 4.1. Our training objective combines the standard diffusion loss Eq. (1) with an attention alignment loss Eq. (5). We extract attention maps from the U-Net decoder at resolutions 8×8, 16×16, 32×32 and supervise them with $M_{grth}$ using Eq. (5). Training uses AdamW (lr=1e-5) for 5k iterations on a single A100 GPU, with batch size of 32 and batch composition of 50% freehand and 50% synthetic sketches.

We use *t2i-adapter-sketch-sdxl-1.0* Lab (2023) pre-trained model for the fine-tuning of the T2I adapter. We apply the same attention supervision strategy by extracting cross-attention maps from decoder layers at 8×8, 16×16, 32×32 resolutions. We use identical hyperparameters (AdamW, lr=1e-5, batch size 32) and train for 5k iterations.

# C SELECTING PARAMETERS IN CONTROLNET AND T2I

As mentioned in Sec. 4.4.1, ControlNet Zhang et al. (2023) and T2I-Adapter Mou et al. (2023) include control parameters that balance sketch fidelity and image realism. ControlNet supports setting a layer-dependent scale factor $w$ for the conditional branch connections. In practice, it is sufficient to define $w$ for the top layer, as values for lower layers are automatically inferred based on the layer's depth.

For T2I-Adapter, we experiment with two parameters Jiang et al. (2025): $s$, which controls how much the conditional features are added in residual connections, and $\tau$, which controls for which time steps $t > T(1 - \tau)$ the spatial guidance is added, where $T$ is the total number of steps used.

## C.1 SELECTING THE BEST SETTING

To select the best settings of control parameters, we run a user study, where for each of the baselines we ask a user to choose among a few possible image options obtained with different parameter settings. 7 participants took part in this user study. The results of the user study are summarized in Fig. 12. Visual results are shown in Figs. 13 to 16, while numerical results are shown in Tab. 5.

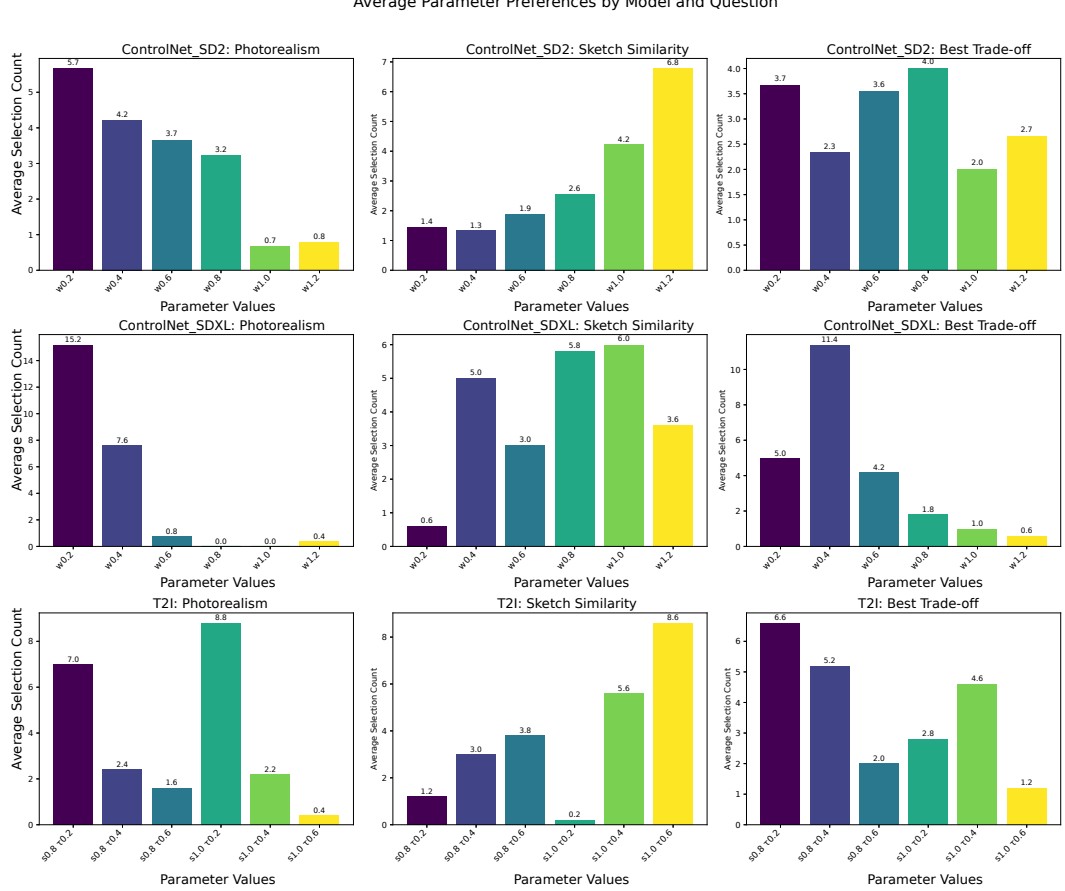

Figure 12: Parameter preferences by model by question.

| Baseline | Param | FID $\downarrow$ | $CLIP_{sim} \uparrow$ | LPIPS $\downarrow$ |
|---|---|---|---|---|
| ControlNet SD2 | w0.2 | 141.46 | 0.911 | 0.780 |
| | w0.4 | 138.62 | 0.927 | 0.774 |
| | **s0.6** | 135.60 | 1.136 | 0.773 |
| | w0.8 | 137.35 | 0.765 | 0.773 |
| | w1.0 | 143.42 | 0.668 | 0.786 |
| | w1.2 | 147.53 | 0.468 | 0.791 |
| ControlNet SDXL | w0.2 | 162.35 | 0.032 | 0.822 |
| | **w0.4** | 174.46 | 0.027 | 0.825 |
| | w0.6 | 196.35 | -0.412 | 0.830 |
| | w0.8 | 236.58 | -0.455 | 0.846 |
| | w1.0 | 219.09 | -0.222 | 0.836 |
| | w1.2 | 243.59 | -0.039 | 0.831 |
| ControlNext SDXL | w0.2 | 131.58 | 0.841 | 0.772 |
| | **w0.4** | 134.09 | 0.909 | 0.774 |
| | w0.6 | 149.19 | 0.279 | 0.781 |
| | w0.8 | 192.92 | -0.357 | 0.796 |
| | w1.0 | 232.27 | -1.036 | 0.843 |
| | w1.2 | 237.38 | -0.934 | 0.837 |
| T2I | **s0.8 $\tau$ 0.2** | 144.33 | -0.203 | 0.813 |
| | **s0.8 $\tau$ 0.4** | 159.82 | 0.213 | 0.819 |
| | s0.8 $\tau$ 0.6 | 165.52 | 0.057 | 0.827 |
| | s1.0 $\tau$ 0.2 | 149.13 | 0.132 | 0.821 |
| | s1.0 $\tau$ 0.4 | 152.41 | 0.213 | 0.820 |
| | s1.0 $\tau$ 0.6 | 144.05 | 0.525 | 0.809 |

Table 5: Detailed performance comparison of all baseline models with different parameters. The settings highlighted in bold are the versions selected for the main paper.

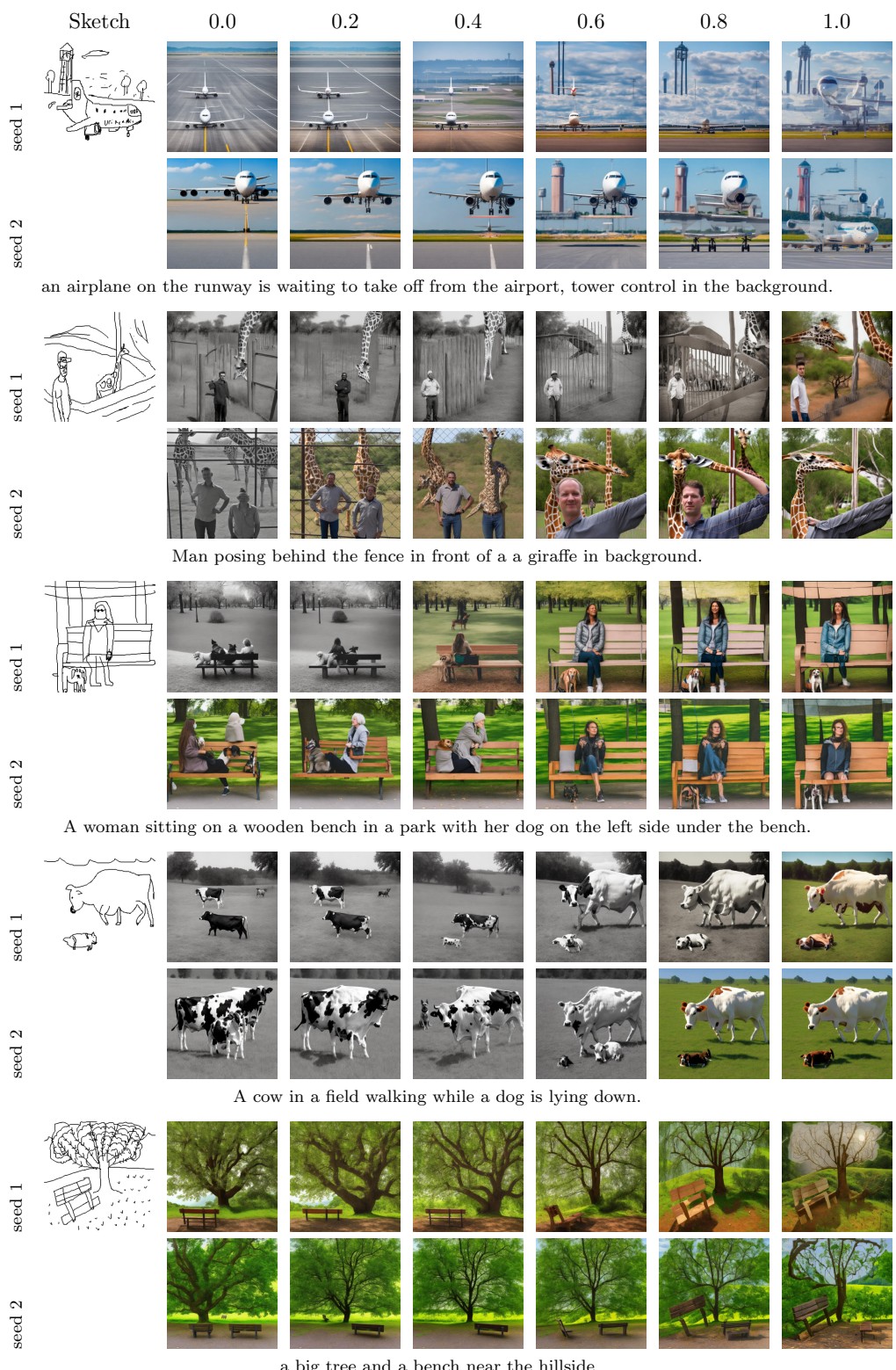

Figure 13: Visualization of image generated with ControlNet SD2 for different values of control parameter. Each row shows results for a specific sketch and text prompt using two random seeds. Text captions are shown below each sketch group.

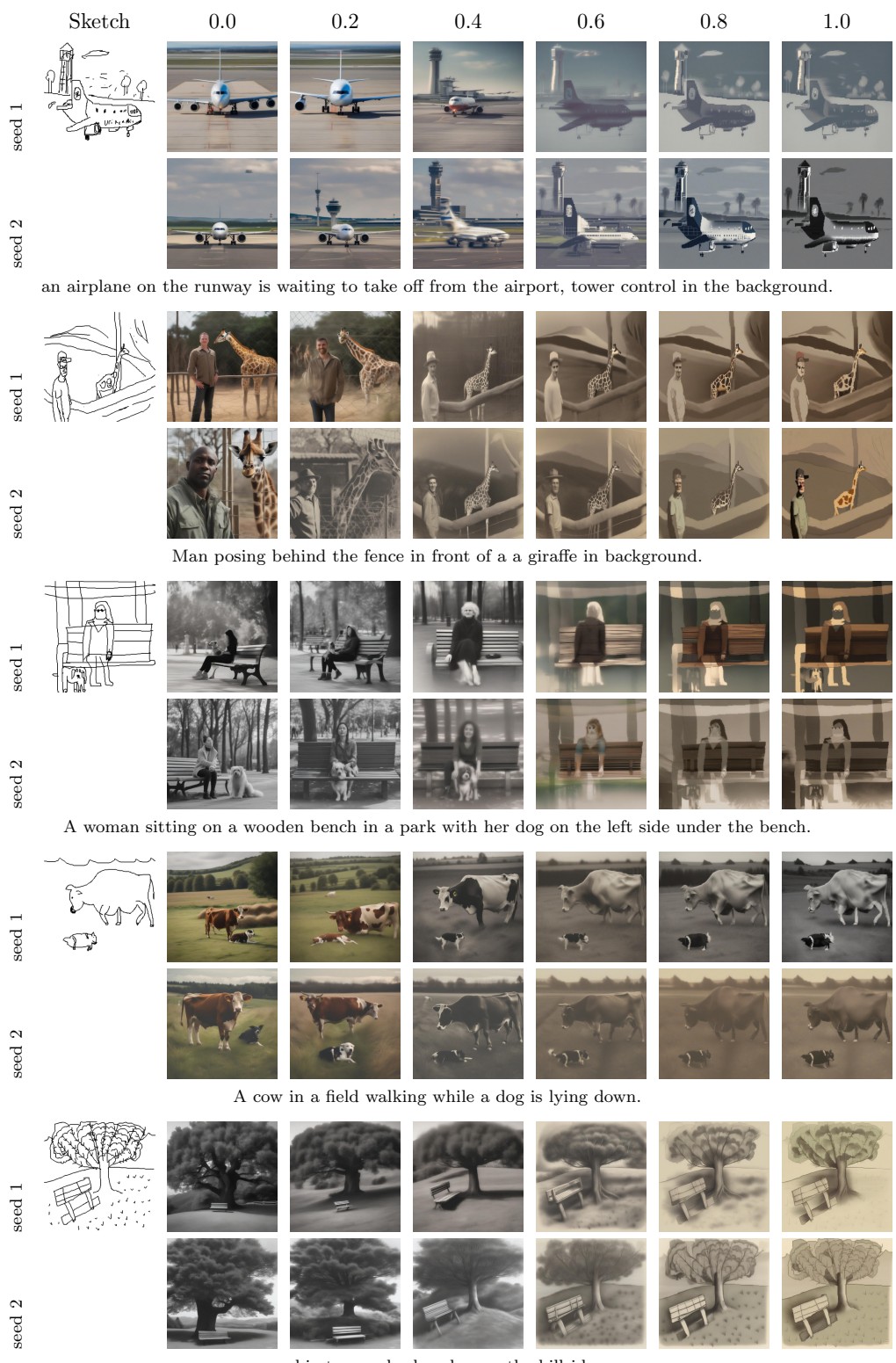

Figure 14: Visualization of image generated with ControlNet SDXL for different values of control parameter. Each row shows results for a specific sketch and text prompt using two random seeds. Text captions are shown below each sketch group.

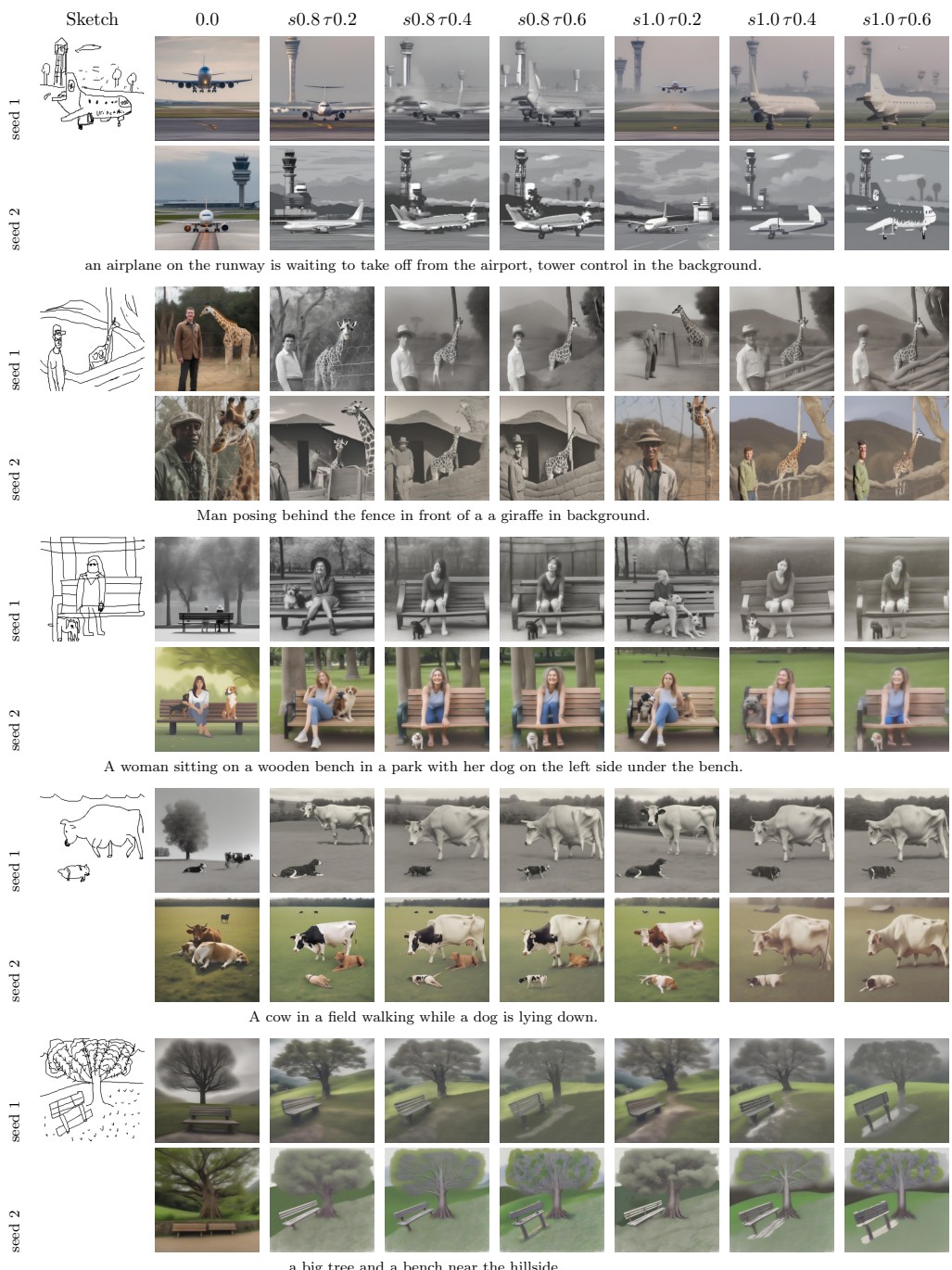

Figure 15: Visualization of image generated with T2I with SDXL model for different values of control parameters. Each row shows results for a specific sketch and text prompt, generated using two random seeds. Text captions are shown below each sketch group.

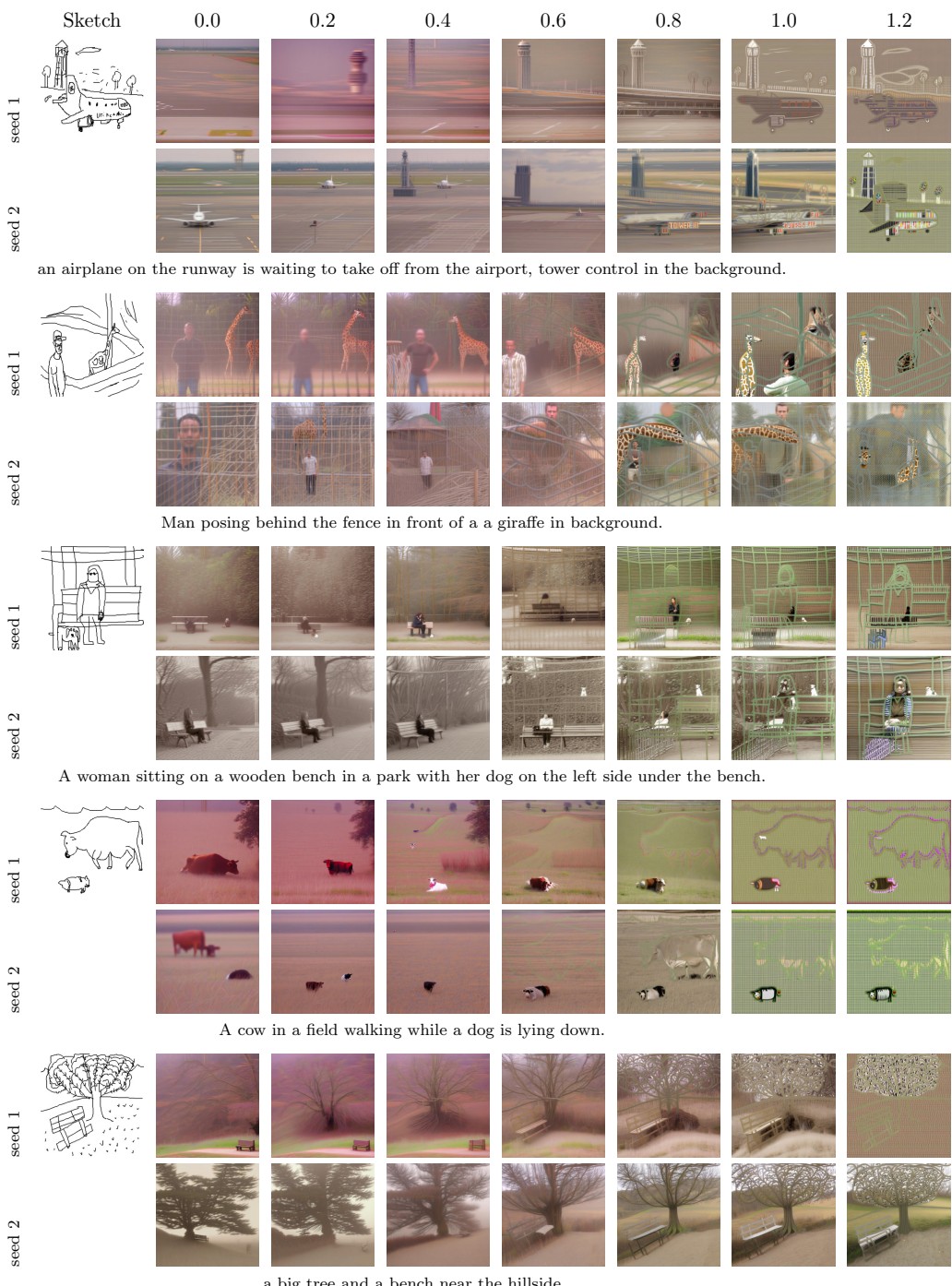

Figure 16: Visualization of image generated with ControlNext for different values of control parameter. Each row shows results for a specific sketch and text prompt using two random seeds. Text captions are shown below each sketch group.

# D ADDITIONAL ABLATIONS

## D.1 TRAINING ON FREEHAND SKETCHES ONLY

To evaluate the importance of training on both synthetic and freehand sketches, we train a variant of our model using only freehand sketches optimized with the full objective in Eq. (6). As shown in Tab. 6, it results in worse performance on freehand sketches across all metrics.

| Method | FID ↓ | CLIP ↑ | LPIPS ↓ |
|---|---|---|---|
| Freehand only | 142.27 | 0.955 | 0.750 |
| Ours (full) | **121.973** | **1.291** | **0.739** |

Table 6: Ablation: Comparing training on freehand sketches alone versus training on a combination of freehand and synthetic sketches, with evaluation performed on freehand sketches from the FS-COCO dataset Chowdhury et al. (2022).

## D.2 TRAINING ON SYNTHETIC SKETCHES ONLY

We additionally investigate a variant trained exclusively on synthetic sketches, denoted as *Ours (synthetic)*. This ablation allows us to evaluate whether our architecture can generalize when restricted to sketches with pixel-aligned ground-truth images.

### D.2.1 PERFORMANCE ON SYNTHETIC EDGE-ALIGNED SKETCHES

As shown in Fig. 17, *Ours (synthetic)* achieves performance comparable to state-of-the-art edge-conditioned methods such as ControlNet (Zhang et al., 2023) and T2I-Adapter (Mou et al., 2023). This indicates that our modulation network and attention-based supervision are effective in capturing the sketch structure when constrained to just edge maps.

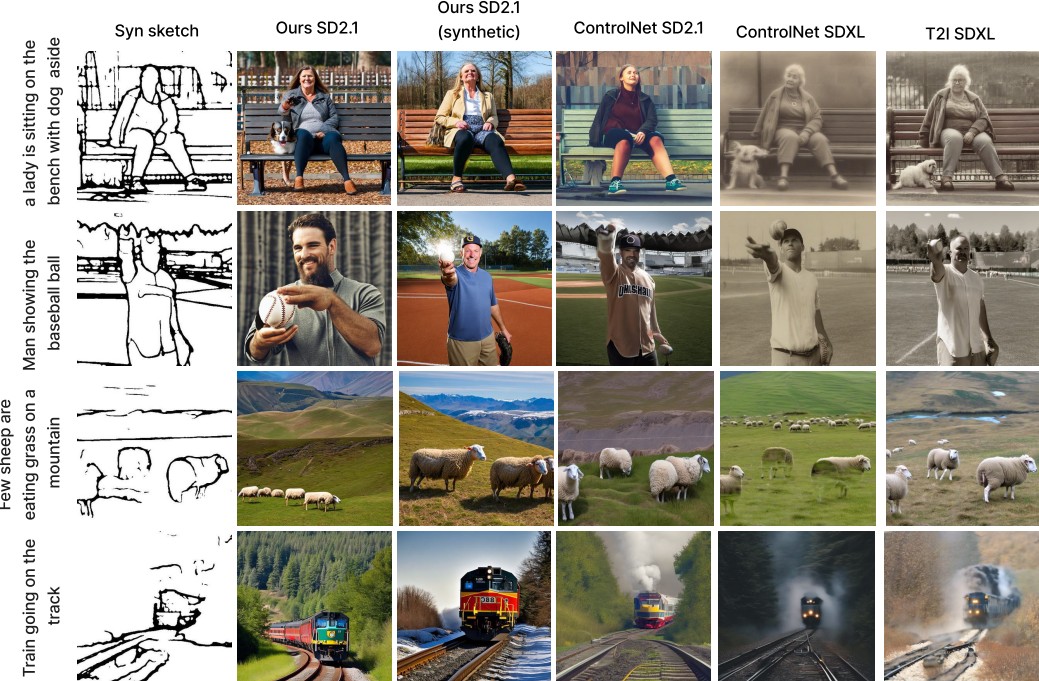

Figure 17: Qualitative results comparison between our method, our method trained on only synthetic sketches, Controlnet, and T2I adapter

Quantitatively, our synthetic variant outperforms both ControlNet SD2.1 and SDXL across all metrics, achieving lower FID (106.74 vs. 124.57 / 112.69) and LPIPS (0.423 vs. 0.458 / 0.432), as

well as higher CLIP similarity (1.280 vs. 1.247 / 1.286). While T2I-Adapter SDXL achieves the best overall scores, our method remains competitive, particularly in FID and perceptual similarity, demonstrating the effectiveness of our approach using only synthetic sketches.

| Method | FID ↓ | CLIP ↑ | LPIPS ↓ |
|---|---|---|---|
| ControlNet SD2.1 | 124.57 | 1.247 | 0.458 |
| ControlNet SDXL | 112.69 | 1.286 | 0.432 |
| T2I-Adapter SDXL | **102.43** | **1.383** | **0.379** |
| Ours (synthetic) | 106.74 | 1.280 | 0.423 |
| Ours (full) | 109.08 | 1.193 | 0.484 |

Table 7: Quantitative comparison of our method trained on synthetic sketches (*Ours (synthetic)*) against baselines trained similarly, evaluated on synthetic sketches, as the ones shown in Fig. 17. The best results are highlighted in bold, and the second-best results are underlined.

### D.2.2 PERFORMANCE ON FREEHAND SKETCHES

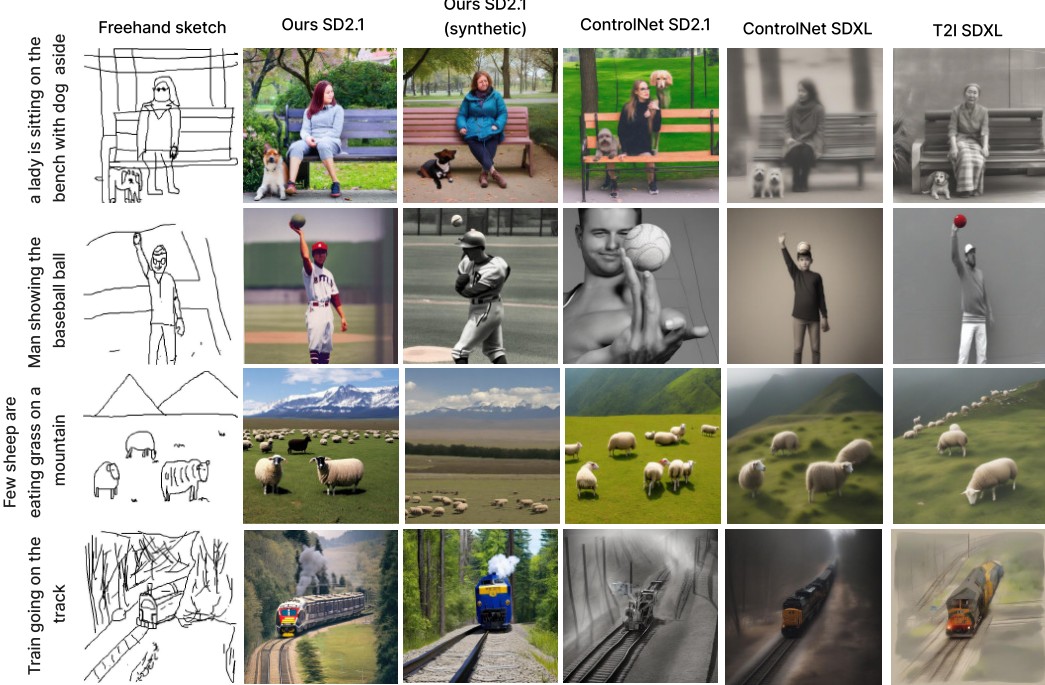

Figure 18: Qualitative comparison between our method, our method trained on only synthetic sketches, ControlNet, and T2I Adapter tested on freehand sketches.

We further evaluate this model on freehand sketches to test its robustness beyond pixel-aligned inputs. As summarized in Tab. 8 and illustrated in Fig. 18, training solely on synthetic sketches leads to a clear drop in performance compared to our full model: CLIP similarity falls to 0.817 and LPIPS increases to 0.793. This suggests that while synthetic training suffices for edge-like inputs, exposure to genuine freehand sketches is essential to handle abstraction and distortion reliably. The qualitative results in Fig. 18 confirm this gap: *Ours (synthetic)* often captures coarse scene layout but struggles with object details and proportions, whereas the full model better respects both semantics and realism.

### D.3 NUMBER OF NOISE STEPS

We first evaluate the noise steps on which our modulation network is trained. We experiment with 10%, 20%, 30%, 40% and 50% of timesteps corresponding to high noise regimes. However, as can

| Method | FID ↓ | CLIP ↑ | LPIPS ↓ |
|---|---|---|---|
| ControlNet SD2.1 | 135.59 | 1.136 | 0.773 |
| ControlNet SDXL | 174.46 | 0.027 | 0.825 |
| T2I-Adapter SDXL | 159.82 | 0.213 | 0.819 |
| Ours (synthetic) | 132.17 | 0.817 | 0.793 |
| Ours (full) | 121.97 | 1.291 | 0.739 |

Table 8: Quantitative comparison of our method trained on synthetic sketches only (*Ours (synthetic)*) with edge-conditioned baselines tested on freehand sketches

be seen in Fig. 19, as we train on more time steps, both FID and CLIP scores gradually degrade, over-constraining the model. A similar observation was reported in the T2I-Adapter paper Mou et al. (2023), which also emphasized focusing training on high-noise regimes. Our choice of restricting training to the top 10% of noise steps not only improves performance but also leads to faster convergence compared to training on a broader noise range. The rest of the experiments in this section are trained for 10% of the highest noise steps.

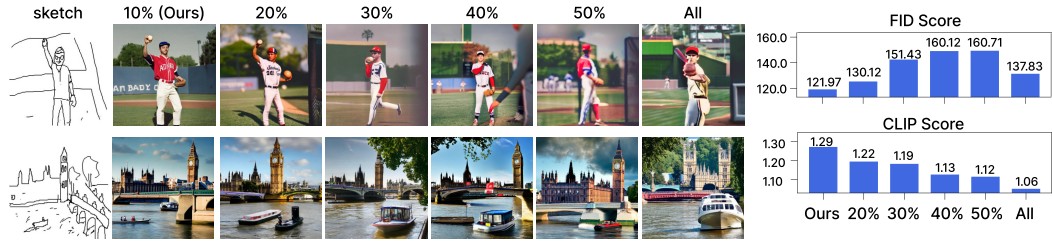

Figure 19: Qualitative and quantitative evaluation when training with 10%, 20%, 30%, 40%, 50% and 100% of timesteps corresponding to high noise regimes. Modulation is performed for the same number of time steps as the respective training. Please refer to Sec. D.3 for the detailed discussion.

## D.4 CONTROLNET AND OUR MODULATION NETWORK

We experimented with integrating our modulation network into ControlNet. As shown in Tab. 9, this combination yields only marginal performance changes, likely due to the dominant influence of ControlNet's conditional branch.

| Method | Setup | FID↓ | CLIP↑ | LPIPS↓ |
|---|---|---|---|---|
| | Zero-shot | 135.595 | 1.136 | 0.773 |
| | $\mathcal{L}_{\text{noise}}$ only (syn) | 136.821 | 1.141 | 0.771 |
| CntrlNet SD2.1 (Zhang et al., 2023) | $\mathcal{L}_{\text{noise}}$ only (free) | 139.872 | 1.042 | 0.789 |
| | $\mathcal{L}_{\text{noise}} + \mathcal{L}_{\text{attn}}$ | 135.891 | 1.196 | 0.768 |
| | $\mathcal{L}_{\text{noise}} + \mathcal{L}_{\text{attn}} + \text{Mod.}$ | 134.771 | 1.192 | 0.767 |
| **Ours** | Full | **121.973** | **1.291** | **0.739** |

Table 9: Comparison on 475 test sketches from the FS-COCO dataset (Chowdhury et al., 2022) with ControlNet versions.

