# OpenReview forum: "SketchingReality: From Freehand Scene Sketches to Photorealistic Images"
_ICLR.cc/2026/Conference — ICLR 2026 Poster_

### Official Review · Reviewer_2bmT · 2025-10-29

**Soundness:** 2
**Presentation:** 2
**Contribution:** 1
**Rating:** 2
**Confidence:** 4

**Summary:**

### Summary

This paper introduces a method for scene sketch-to-photo synthesis. The paper argues that the previous study mainly focuses on the conversion from edge maps to photos. Thus, the proposed method is developed to convert actual freehand sketches to photorealistic images, leveraging semantic sketch features. Compared to simple ``sketch-to-photo'' methods like ControlNet and T2I Adapter, the proposed method yields more plausible results.

**Strengths:**

The proposed method yields better results compred to *standard* image-to-image conversion methods like ControlNet and T2I Adapter.

**Weaknesses:**

### Major comments regarding the novelty

Although the paper claims the majority of prior methods just convert edge maps to photos (this may be true to some extent; for example, ControlNet is often trained on edge maps), the conversion from abstract freehand scene sketches to photos is actively studied, as in but not limited to [a-c].

Compared to those (bunch of) existing attempts, the novelty of the submitted paper is unclear. The novelty and merit (over methods inputting freehand sketches) are not adequately discussed in the related work sections, nor are they compared in the experiments.

- [a] Xiang, Xiaoyu, et al. "Adversarial open domain adaptation for sketch-to-photo synthesis." WACV 2022.
- [b] Wang, Jiayun, et al. "Unsupervised scene sketch to photo synthesis." ECCV 2022.
- [c] Koley, Subhadeep, et al. "Picture that sketch: Photorealistic image generation from abstract sketches." CVPR 2023.

**Questions:**

I would ask the authors to clarify the novelty and merit compared to existing methods for converting free-hand (or abstract) sketches to photos.

**Details Of Ethics Concerns:**

n.a.

---

> ### Author Response · Authors · 2025-11-25
>
> Thank you for your feedback.
>
> The three suggested papers [a–c] are not state-of-the-art; those are early GAN-based works. As noted in lines [107–109] of our paper, numerous recent studies have shown that diffusion models substantially outperform GAN-based approaches. For this reason, we focused our discussion and comparisons on more recent and advanced diffusion-based methods.
>
> The main limitation of the cited works is their restricted generalization: they are trained on narrow, domain-specific data and do not extend beyond those domains. Their primary challenge is handling minor variations in sketch style, rather than supporting open-vocabulary scenarios with diverse layouts and object categories. In contrast, a key objective of our method is to leverage foundational models that generalize at test time to a wide range of unseen cases, including varied scene layouts, object categories, and text prompts, while also addressing the types of abstraction common in freehand sketches on an object and scene levels (e.g., representing a mass of trees with only a few abstractly drawn individual trees), incoherent relative object sizes, and distorted perspective.
>
>
> Additionally, two of the cited papers [a, c] handle only single-object sketches, while [b] operates on edge-map-style inputs with variation primarily in line style or hatching.
>
> More specifically:
>
> [a] is trained on single-object sketches conditioned on class labels and supports 24 categories. It does not handle scene sketches or text prompts, and it produces only object cutouts rather than holistic images.
>
> [b] is a GAN-based predecessor to ControlNet-style approaches and focuses on transferring edge-map sketches to images with strong spatial alignment. It does not generalize beyond its training domain. Its sketch style robustness relies on preprocessing sketches with an edge detector --- an orthogonal technique that could complement our method. In contrast, our approach explicitly tackles abstraction, inconsistent object scales, and perspective distortions common in freehand scenes.
>
> [c] is trained on single-object sketches and supports only three object categories (chairs, handbags, and shoes).
>
> We hope this clarifies the substantial differences in scope, technical approach, and generalization abilities between these prior works and our proposed method.

---

> > ### Comment · Reviewer_2bmT · 2025-11-28
> >
> > Thanks for the response.
> >
> > I agree that the above literature is outdated (in its use of GANs); I apologize for including the older examples. Sketch-to-photo is also an active area in the diffusion era, in addition to (relatively old) ControlNet, which is mainly used for "edge-map"-to-photo synthesis. From my view, actually, many diffusion-related works input "real" sketches to synthesize photos.
> >
> > Following are just a few examples: An early attempt is Voynov's SIGGRAPH '23 work [d], which already deals with the sketch-to-photo synthesis with high-level abstraction. Newer examples [e,f] and many other papers by Kolay et al. also incorporate "real" sketches to generate photos, including abstract ones. While these methods do not encompass the entire scene, there are also methods targeting sketch-to-scene photo generation using diffusion models, for example [g,h].
> >
> > - [d] Voynov, A., Aberman, K., & Cohen-Or, D. Sketch-guided text-to-image diffusion models. SIGGRAPH 2023.
> > - [e] Koley, Subhadeep, et al. "It's All About Your Sketch: Democratising Sketch Control in Diffusion Models." CVPR 2024.
> > - [f] Koley, Subhadeep, et al. "Text-to-image diffusion models are great sketch-photo matchmakers." CVPR 2024.
> > - [g] Zhang, Tianyu, et al. "Sketch-guided scene image generation with diffusion model." Computers & Graphics, 2025.
> > - [h] Wu, Zhenbei, et al. "Sketchscene: Scene sketch to image generation with diffusion models." ICME 2023.
> >
> > I understand the paper already refers to many of them above in the related work section. Meanwhile, I think many works actually correctly state the "sketches", unlike the paper's argument:
> > >Previous literature has largely focused on edge maps, often misnamed “sketches”, yet algorithms that effectively handle true freehand sketches.
> >
> > A potential explanation may be that the majority of existing methods are trained on edge maps, whereas the proposed method is developed to be trained on real sketches, such as FS-COCO. (By the way, [h] is actually trained with FS-COCO).
> >
> > However, the main assumption by existing "sketch-to-photo" methods using edge-to-photo-trained diffusion models is that, thanks to the strong diffusion prior, they exhibit certain generalizability to input real, free-hand sketches. The realism-vs-pixel-alignment tradeoff is a classical issue that has been discussed in earlier examples [d].
> >
> > If the submitted work aims to argue against the above assumption, I think the paper should include thorough comparisons to state-of-the-art sketch-to-photo synthesis methods (other than ControlNet).
> >
> > ---
> >
> > Overall, I remain concerned that I may overlook the important contribution of this work. I would appreciate it if authors (or other reviewers, who tend to rate relatively high scores) could clarify the key, fundamental technical contribution of the work. I do not want to misjudge this paper.

---

> ### Author Response · Authors · 2025-12-01
>
> Thank you for your comments. We identified the following concerns in this response that we address below:
>
> (1) *Concerns regarding our argument that methods trained primarily on synthetic, edge-based sketches struggle to generalize effectively to freehand sketches. The reviewer cites example [d] as evidence suggesting that such generalization may be achievable. We address it below.*
>
> **First**, we would like to clarify that our position is not unique: as the reviewer correctly notes, prior work such as [e] also explicitly reports that models trained on synthetic sketches fail to generalize well to real, freehand sketches. However, [e] focuses on the considerably simpler setting of single-object sketch synthesis, whereas our work targets the more challenging problem of generating photorealistic images from abstract, multi-object scene sketches.
>
> A direct comparison with [e] is unfortunately not feasible, as the authors confirmed to us that their implementation is no longer available: the code was lost due to hardware failure. They also shared that they had attempted to extend their method to scene sketches but were unsuccessful. This limitation may stem from two factors:
>
> (a) their reliance on projecting sketch features into text-feature space, which may not provide sufficient representational capacity for complex scene-level sketches, and
>
> (b) their dependence on a CLIP-based training loss, which can struggle to enforce fine-grained geometric alignment for scene-level constraints. We also note that CLIP-loss training requires decoding images at every iteration, making it substantially slower compared to our attention-based loss formulation.
>
> However, we still made every effort to compare against their approach (referred to as AboutYourSketch). Since the original code is unavailable, we evaluated our method on the example images provided in their paper. Please see Fig. 10 in the revised appendix, with Sec. A6 offering a detailed discussion of the differences in task formulation, training data, and model behavior. As shown in Fig. 10, our method not only generalizes well to their setting but also achieves performance comparable to their specialized single-object approach, despite not being trained on single-object sketches and being designed for a significantly more general scene-level task.
>
> **Second**, we have provided a detailed comparison with [d] Voynov et al. (SIGGRAPH '23) in the original submission, denoted as SG in our paper (L329). While the code is not publicly released, the authors kindly ran their method on our test set. **Table 1 includes the numerical comparison, and Figure 3 shows the visual comparison, demonstrating that their method does not generalize to freehand scene sketches**. In their paper, they only show a limited number of examples on single-object sketches.

---

> ### Author Response · Authors · 2025-12-01
>
> (2) *There is concern that we have not compared our approach against a sufficiently broad set of state-of-the-art methods to substantiate the claim that existing techniques do not adequately address the task we investigate --- namely, generating photorealistic images from abstract freehand scene sketches --- and that our proposed method represents a significant contribution by offering a superior solution to this problem.*
>
> We have already discussed our comparison with works [d] and [e] above. We would like to clarify that [f] is a retrieval-based approach rather than a generative method, and therefore is not relevant to our problem. The method [g] by Zhang et al. (Computers & Graphics 2025) is covered in L148--158 (Related Work); however, the authors do not release their code, making a comparison impossible under ICLR rules. Conceptually, their approach has several important limitations relative to ours: (i) it introduces substantial inference-time computational overhead, and (ii) as shown in Figure 5 of their paper, the method may produce outputs that lack realism or fail to reflect the intended object arrangement depending on parameter choices, indicating difficulty achieving a favorable trade-off between photorealism and sketch adherence.
>
> Similarly, [h] Wu et al. (ICME 2023, SketchScene) is discussed in lines 145--148. This is an early diffusion-based approach trained on FS-COCO in image space, preceding latent diffusion models. The authors confirmed to us that the code is not available, again preventing a comparison under ICLR rules. Based on their published results, the visual quality appears substantially below that of modern latent diffusion methods, including ours.
>
> To further strengthen our evaluation, we additionally compared against a powerful, general-purpose state-of-the-art editing model, Flux Kontext. Although not explicitly trained for sketch-to-image generation, it benefits from extremely large-scale training. We prompted it with "Convert a sketch to a photorealistic image of ...". Visual comparisons are provided in Appendix Fig. 9, with further discussion in Sec. A5. Quantitatively, our method achieves a lower FID (123.595 vs. 132.487) and a higher CLIP sketch--image similarity score (1.272 vs. 1.226), demonstrating both improved realism and stronger sketch alignment. Even with extensive pretraining, Flux Kontext underperforms relative to our approach.
>
> In summary, of the papers mentioned, we are either already comparing to them (d, e), or it is impossible to compare as there is no source code (g,h), or the paper is a retrieval, not a generative method (f).
>
> (3) *A request for clearer articulation of the fundamental technical contribution of our work.*
>
> We appreciate the opportunity to re-emphasize this. Our contributions center on two key innovations:
>
> (i) A modulation network that integrates semantic sketch-encoder features into the generative diffusion process. This design enables the model to effectively leverage high-level, freehand sketch semantics --- crucial for handling abstract, multi-object scene sketches --- and leads to significantly improved alignment and generalization compared to relying solely on VAE-based or text-projected representations.
>
> (ii) A cross-attention supervision mechanism that provides direct, fine-grained alignment between sketch regions and generated image content. This allows efficient and stable training on freehand sketches.
>
> Together, these two components form the technical foundation that enables our method to operate robustly in a setting that has been challenging for existing approaches.

---

### Official Review · Reviewer_JH8r · 2025-10-30

**Soundness:** 3
**Presentation:** 3
**Contribution:** 2
**Rating:** 6
**Confidence:** 3

**Summary:**

The present work seeks to improve sketch-guided image generation algorithms, e.g.,ControlNet and Adapter-based models. Specifically, the authors leverage an existing sketch-focused semantic embedding model to generate features that help guide the training of a new model that includes a novel modulation network trained on a loss that estimates likelihoods of different sketch pixels belonging to different semantic categories. Notably, the authors train on a mix of automatically generated sketches from images, and freehand sketches, along with language supervision leading to the ability to represent sketches and their relationship to images at multiple levels of abstraction.
I generally found the results quite compelling, although the improvement in metrics did not seem to always seem to qualitatively lead to images that looked much better. I found that while individual sections were well written, the organization of sections was somewhat disorganized requiring much back and forth between sections of the paper to understand the key methods and results. I elaborate more in the sections below.

**Strengths:**

* Both theoretically and empirically, I found the attention supervision loss to be quite elegant and effective. I find this to be among the paper’s stronger contributions and a key step towards translating freehand sketches to semantically coherent images.
* As with any good vision paper, I found the ablation experiments and comparisons to existing methods quite thorough and generally compelling.
* While I do have issues with the general clarity of the methods overall, I found some of the overview sections (e.g., preliminaries on LDM) well executed for the amount of space taken up by the section.

**Weaknesses:**

* The paper is at times hard to read because of its structure. For example, the modulation network is introduced before the reader knows what the sketch features that it uses are. I recommend laying out the necessary details of all the ‘ingredients’ of a module before diving into details about the module.
* I think there needs to be more details of the user study in the main text. I also find 23 participants to be quite a small pool.
* While I do find the interleaved discussion of the results in the Ablation section well written, the conclusion is quite limited both in terms of length and overall takeaways.
* While the modulation network and the attention supervision approaches are key contributions, the gains (here measured via improvements in FID, CLIP, and LPIPS) are fairly incremental with the image quality not appearing to be vastly superior. I think showing that users vastly prefer using this model relative to others would be compelling.

**Questions:**

* Perhaps because I’m not deeply familiar with Ham et al., 2023, but while I understand why low variance is penalized in the scale and shift maps (S, B), could the authors motivate the L1 regularization of these feature maps too? Is this just to smoothen the training process or is there a deeper theoretical significance?
* Could more be said about how the last 3 layers of the sketch encoder (from Bourouis et al.) was fine-tuned?
* I think currently the ‘Attention Supervision’ part of the training pipeline is the least clear, particularly Eq 5. I would recommend a greater explanation of the Sun et al. formulation either in that section or in the Appendix.
* I would like to better understand why training on a mix of synthetic and freehand sketches led to more vivid color palettes as the authors note.

---

> ### Author Response · Authors · 2025-11-25
>
> Thank you for your thorough feedback!
> Below, we address the concerns mentioned in the Weaknesses section and the raised questions.
>
> ## User preference for our model and the number of participants
> Our study shows a consistent preference for our method over the other approaches (Table 2 in the paper).
>
> The observed differences are statistically significant. For example, in the best trade-off category, all pairwise comparisons yield p-values < 1e-6, indicating that our approach achieves a superior balance of photorealism and sketch similarity according to user preference. Given these statistically significant results, the sample of 23 participants is enough.
>
> Across all baselines, our method is consistently preferred over the competing approaches listed in the leftmost column. For example, when compared to SDXL T2I with s=0.8, \tau=0.2, our method was chosen by users 66.0% of the time, versus 20.6% for the base method and 13.4% for undecided votes. Similarly, for ControlNet SD2, users favored our method 88.8% of the time.
>
> ## The L1 loss
> It promotes minimal changes to the predicted noise, achieving a balance between respecting the conditioning and maintaining the expected noise distribution.
>
>
> ## Sketch encoder finetuning details
> The sketch encoder [1] is based on a dual-path CLIP-ViT architecture with 12 transformer blocks.
> We fine-tune all weights of the last three blocks of both paths, including all submodules (LayerNorm, Multi-Head Self-Attention (MHSA)/v-v attention, and feed-forward layers). This strategy preserves semantic understanding while enhancing feature extraction for the modulation network. We will publicly release our training and inference code.
>
> [1] Ahmed Bourouis, Judith E Fan, and Yulia Gryaditskaya. Open vocabulary semantic scene sketch
> Understanding. CVPR, 2024.
>
>
> ## The attention supervision loss (Eq. 5)
> The attention supervision loss in Eq. (5) follows the formulation introduced by Sun et al. (2024), originally developed for layout-based image generation. Their key idea is to guide the model’s cross-attention maps so that each text token attends primarily to the spatial region of the image corresponding to its semantic entity. In our setting, the same principle applies: each sketch object P_i has a corresponding ground-truth region in the sketch-derived attention map M_{grth}, and the model’s cross-attention maps M are encouraged to focus on that region.
>
> The ratio in the first term is the fraction of total attention for object P_i  that lies within its ground-truth spatial region b_i  (binary mask), derived from M_grth. A value close to 1 indicates that little is leaking outside the area b_i. The second term, weighted by \lamda_reg, is a regularization that further discourages attention from spilling outside b_i. It sums the attention values on pixels outside the target region.
> Summing over layers \gamma aggregates contributions across multiple cross-attention layers, ensuring that alignment is encouraged throughout the network rather than in a single layer.
>
>
> ## Fine-tuning baselines leads to more vivid colors when tested on freehand sketches compared to the zero-shot setting
>
> The main reason for this is that our fine-tuning strategy with cross-attention alignment allows the model to generalize to freehand sketches at inference time. Fine-tuning on a mix of freehand sketches and synthetic helps the models to preserve their previously learned priors, as we can efficiently use L_noise only for synthetic sketches.

---

### Official Review · Reviewer_L6kH · 2025-11-01

**Soundness:** 2
**Presentation:** 2
**Contribution:** 2
**Rating:** 4
**Confidence:** 4

**Summary:**

The paper tries to tackle generating photorealistic images from freehand scene sketches and preserving structure and semantics. It augments a pretrained text-to-image model Stable Diffusion 2.1 with a lightweight modulation network that scales and shifts the predicted noise using sketch-conditioned features. A cross-attention supervision loss enforces text–spatial alignment. Training mixes human and synthetic sketches and emphasizes early high-noise timesteps to balance cost and global layout learning. Experiments show consistent gains over baselines on FID, CLIP similarity, and LPIPS.

**Strengths:**

The method handles abstract and deformable sketches for the scene-level sketch-to-photo generation. Attention supervision explicitly ties language tokens to spatial regions and the modulation head is lightweight and only active in early timesteps, which keeps computation modest. It plugs into a standard SD2.1 pipeline without re-architecting. This modularity makes the approach easy to reproduce.

**Weaknesses:**

The paper does not offer a mechanistic explanation or theoritical analysis for why the noise-modulation head works. Evidence is largely empirical like metric tables and ablations without probing the internal reasons. Moreover, it does not clearly position the method against mainstream fine-tuning techniques such as LoRA comparison and the core distinction from other fine-tuning techniques remains under-analyzed.

The paper does not clearly articulate the mechanistic between the noise-modulation head and the target attention maps like why cross-attention alignment must be enforced via scale/shift on the predicted noise. The necessity of using target attention maps through this specific noise-space finetuning pathway remains not solid and reasonable.

The backbone choice is not state-of-the-art as the method is demonstrated on Stable Diffusion 2.1 only. There are no experiments on newer text-to-image backbones (like SDXL or FLUX). Without such results, generality are weakened and the figures presented seem relative lower quality in current phase, hurting the claim of the method effectiveness.

**Questions:**

Could the author explain more about why must cross-attention alignment be enforced via scale/shift on predicted noise finetuning?

---

> ### Author Response · Authors · 2025-11-25
>
> Thank you for your thorough feedback!
> Below, we address the concerns mentioned in the Weaknesses section and the raised questions.
>
> ## Relation between the modulation network and cross-attention alignment
> As noted in the introduction (lines 100–102), our method’s strong performance arises from two main mechanisms:
>
>  (ii) the use of semantic sketch features via integration with the proposed modulation network in latent space, and
>
>  (iii) enabling efficient training on freehand sketches through a loss function that emphasizes the semantic structure of the sketch inputs.
>
> Using the modulation network for conditioning and the cross-attention alignment loss are complementary mechanisms, not directly tied to each other.
>
> As shown in the paper, cross-attention alignment can also be applied independently of the modulation network. Specifically, we use this loss to adapt other baselines to freehand sketches. Importantly, the issue is not whether LoRA or full-weight fine-tuning is used; rather, it concerns the choice of loss. Fine-tuning on freehand sketches with standard diffusion-model losses leads to degraded performance, as there is no pixel-level alignment between the conditioning sketch signal and the target ground truth. For instance, fine-tuning ControlNet with the original loss (“L_noise only (free)” in Table 1, third line) results in lower alignment with sketches compared to the zero-shot baseline (first line), with CLIP similarity dropping from 1.136 to 1.042 and weaker FID scores (lines 357–361).
>
> To further evaluate the role of the fine-tuning mechanism, we additionally train ControlNet using the original loss but with LoRA applied in two setups: (i) LoRA applied only to the ControlNet branch, and (ii) LoRA applied to both the ControlNet and main branches. We train on a combination of synthetic and freehand sketches. Table 3 and Figure 8 show results consistent with those reported in the main paper: fine-tuned ControlNet achieves slightly better alignment with the sketches, but the overall image quality deteriorates. Moreover, the fine-tuned ControlNet variants perform substantially worse than our method with either the SD2.1 or SDXL backbones. Detailed discussions are provided in Section A4 in the supplemental.
>
> ## Motivation behind the modulation network
> We use noise modulation to leverage semantic sketch features. Cross-attention alignment loss is independent. For example, ControlNet is inherently limited to VAE sketch features, while T2I adapter features are tied to intermediate layers of the diffusion model. In contrast, our noise modulation network enables efficient use of semantic sketch features. The modulation mechanism itself is well-established [1]; our contribution lies in designing the projection network that maps sketch features to noise modulation parameters, not in the modulation mechanism itself. We will revise the relevant discussion (lines 077-084) to make these points clearer.
>
> [1] Park, T., et al. Semantic Image Synthesis with Spatially-Adaptive Normalization. CVPR, 2019.
>
>
> ## Choice of backbone model in the paper
> We used SD2.1 for all our main experiments due to computational constraints.
>
> To complement the results in the main paper, we implemented our full method with the SDXL backbone, with results reported in supplemental Table 3 and Figure 8. Using a stronger backbone further improves our method across all three metrics: FID, CLIP, and LPIPS (Table 3  \#5 vs. \#4). Switching from SD2.1 to SDXL reduces FID from 123.959 to 117.025, increases CLIP sketch–image similarity from 1.272 to 1.331, and decreases LPIPS from 0.743 to 0.708. Detailed discussions are provided in Section A3 in the supplemental.
>
> Additionally, we evaluate our method against the recent universal image editing model FluxKontext, which is based on FLUX. Although this model is not specifically trained for sketch-to-image conversion, it benefits from training on a massive dataset. We prompted the model with “Convert a sketch to a photorealistic image of …”. Visual results are provided in supplemental Figure 9, and additional discussions are added to Section A5.
>
> Quantitatively, we achieved a lower FID (123.595 vs 132.487) and a higher CLIP sketch–image similarity score (1.272 vs 1.226), validating both the superior realism and the stronger sketch alignment for our method.
>
> Even with extensive pretraining, FluxKontext underperforms compared to our approach. While integrating our conditional mechanism and cross-attention alignment loss with this backbone is left for future work, this experiment demonstrates that our method, despite using a weaker backbone, achieves superior performance for the task of converting freehand sketches to photorealistic images.

---

### Official Review · Reviewer_pKSW · 2025-11-03

**Soundness:** 3
**Presentation:** 3
**Contribution:** 3
**Rating:** 6
**Confidence:** 4

**Summary:**

This work focuses on the task of scene-level sketch-based photorealistic image synthesis. It proposes a new modulation network and a loss function that emphasize the semantic structure of sketch inputs thereby improving generation quality. The authors have conducted experiments on the FS-COCO dataset and compared their method against existing conditional image generation methods, such as ControlNet, T2I Adapter, and FreeControl. The experimental results demonstrate that the proposed method outperforms these baseline methods across various metrics, including FID, CLIP and LPIPS.

**Strengths:**

1. The proposed method achieves impressive visual quality in its generated results, which is further supported by strong quantitative performance.
2. This work addresses a key challenge in sketch-based image generation, the inherent abstraction and ambiguity of sketches, which often leads to distortion in the results of existing methods. The proposed modulation network effectively addresses this issue by emphasizing the semantic structure of sketches. Consequently, the generated images achieve a good balance between visual quality and consistency with the input sketch
3. The paper is well-written and easy to follow.

**Weaknesses:**

1. The authors state that because a sketch can be abstract and ambiguous, their method focuses on extracting its semantic and structure information. In that case, it raise the question of whether an alternative approach, such as performing sketch captioning first and then feeding the resulting text into a standard T2I model (or baseline methods used in this paper), could be viable. A discussion or comparison against such a two-stage pipeline would be a valuable addition.
2. All experiments are conducted on the FS-COCO dataset, which contains relatively high-quality freehand sketches. To better assess the model’s robustness, it would be beneficial to evaluate its performance on datasets with more abstract or simplistic sketches, e.g., QuickDraw? This would provide greater insight into the model's generalization capabilities with respect to sketch abstractness.
3. The paper focuses on photorealistic image synthesis. It would be interesting to see if the proposed method can generalize to other artistic styles, such as cartoons. Including a few examples or a brief discussion on the model's adaptability to different styles would broaden the perceived applicability of the work.

**Questions:**

(1) The authors mentioned that the sketch encoder requires finetuning. Could the authors provide further details on this process?

(2) What are the corresponding text prompts used to generate the examples shown in Fig. 5?

---

> ### Author Response · Authors · 2025-11-25
>
> Thank you for your thorough feedback!
> Below, we address the concerns mentioned in the Weaknesses section and the raised questions.
>
> ## Comparison to a two-stage approach (VLM sketch captioning + T2I/Baseline)
> Thank you for the suggestion. We evaluated two popular vision–language models (VLMs) on the task of captioning freehand sketches from our test set: LLaVA [1] and Qwen2.5-VL [2].
> First, we discovered that both models omitted a substantial portion of the object categories present in the human-annotated sketch captions (e.g., LLaVA missed 47.21% of objects and Qwen2.5-VL missed 41.93%).
> Second, we used the captions generated by LLaVA as input to the SD2.1-ControlNet baseline. This led to even poorer alignment between generated images and input sketches, as measured by CLIP similarity, which dropped from 1.136 to 0.259.
>
> These results indicate that current VLMs still struggle to reliably interpret abstract freehand sketches.
>
> [1] Liu, H., Li, C., Wu, Q. and Lee, Y.J., 2023. Visual instruction tuning. Advances in neural information processing systems, 36, pp.34892-34916. (https://huggingface.co/llava-hf/llava-1.5-7b-hf)
>
> [2] https://huggingface.co/Qwen/Qwen2.5-VL-3B-Instruct
>
>
> ## Generalization to QuickDraw draw sketches
> Please note that the QuickDraw dataset contains only single-object sketches, whereas our primary target modality is freehand scene sketches. To illustrate how our method performs on single-object sketches relative to specialized state-of-the-art approaches, we include a visual comparison with results from AboutYourSketch [3], a recent method specifically designed for single-object sketches and built on the SD1.5 backbone. The newly added Figure 10 in the supplemental shows that our method not only generalizes well to this setting but also achieves performance comparable to the specialized approach, while being substantially more general. We provide additional details in the supplemental Section A6.
> A numerical comparison was not possible because the authors of [3] have not released their code.
>
> [3]  Koley, S., Bhunia, A.K., Sekhri, D., Sain, A., Chowdhury, P.N., Xiang, T. and Song, Y.Z. It's All About Your Sketch: Democratising Sketch Control in Diffusion Models. CVPR, 2024.
>
>
> ## Support for different styles of generated images
>
> Our method inherits the capabilities of the backbone model. The style is typically handled by separate mechanisms that are orthogonal to our conditioning approach.
>
>
> ## Sketch encoder finetuning details
> The sketch encoder [4] is based on a dual-path CLIP-ViT architecture with 12 transformer blocks.
> We fine-tune all weights of the last three blocks of both paths, including all submodules (LayerNorm, Multi-Head Self-Attention (MHSA)/v-v attention, and feed-forward layers). This strategy preserves semantic understanding while enhancing feature extraction for the modulation network. We will publicly release our training and inference code.
>
> [4] Ahmed Bourouis, Judith E Fan, and Yulia Gryaditskaya. Open vocabulary semantic scene sketch
> Understanding. CVPR, 2024.
>
>
> ## Figure 5: used captions
> Thank you! We added the captions for each of the sketches directly to the figure. Please note that captions are taken directly from the FSCOCO dataset, preserving original punctuation, capitalization, and grammar.

---

### Meta-Review · Area_Chair_NE2o · 2026-01-08

**Summary:**

This work focuses scene-level sketch-based photorealistic image synthesis task. It proposes a modulation network and a loss function that emphasize the semantic structure of sketch inputs thereby improving generation quality. The rebuttal substantially addressed most technical questions.The remaining risks are mainly about paper clarity/structure, the depth of “mechanistic” explanation (even if empirical evidence is relatively solid), and the completeness of SOTA comparisons when code is unavailable. Overall, after rebuttal, the balance of evidence favors accept.

**Reviewer Concerns:**

Resolved:

- The authors tested LLaVA/Qwen2.5-VL captioning plus ControlNet and showed severe object misses and large CLIP alignment drops, convincingly ruling out this alternative.

- Added experiments with SDXL and comparisons to FluxKontext demonstrate the method’s effectiveness beyond a single backbone and outperform strong baselines.

- Additional LoRA-ControlNet experiments clarify that the key limitation lies in diffusion losses on freehand sketches, not in LoRA vs full fine-tuning, strengthening the methodological argument.

- Direct quantitative and visual comparisons to Voynov (SIGGRAPH’23) and discussion of diffusion-era sketch methods clarify why many fail to generalize to freehand scene sketches.

- The rebuttal provided missing implementation details (sketch encoder fine-tuning, attention loss, regularization) and addressed user-study concerns with statistically significant results.

Unresolved:

- Some reviewers may still see the contribution as primarily empirical or engineering-driven, lacking deeper theoretical/mechanistic explanation of why modulation works beyond analogy to SPADE.

- Generalization to styles such as cartoon remains largely discussed rather than empirically demonstrated, leaving a gap for skeptical reviewers.

- Mechanistic Explanation of Modulation + Attention
While clarified as complementary, the causal or theoretical reasoning behind why this combination improves alignment is not fully formalized.

- Claims that prior “sketch” methods are actually edge-map methods may need softer wording and a clearer taxonomy to avoid alienating reviewers familiar with diffusion-era sketch control.

- Several reviewers still desire improved paper structure, clearer takeaway statements, and a stronger concluding synthesis to lock in confidence.

**Reviewer Scores:**

Reviewer  pKSW (initial 6):

The rebuttal directly answered each weakness with concrete evidence (two-stage baseline study, QuickDraw discussion + single-object comparison, fine-tuning details, prompts). This reviewer was already positive; full discussion would likely increase confidence and nudge the score upward.

Reviewer  L6kH (initial 4):

The added LoRA experiments, SDXL results, and FluxKontext comparison address major concerns about positioning and backbone generality. If the reviewer’s main blocker is “no mechanistic explanation,” they might still stay at 4; but given the strengthened empirical case and clarified roles of loss vs modulation, a move to a borderline-neutral is plausible.

Reviewer  JH8r (initial 6):

They were already above threshold; rebuttal addresses questions well. Remaining issues are mostly presentation/organization, which typically doesn’t change the score unless severe. Likely stays at 6.

Reviewer  2bmT (initial 2):

Given the reviewer acknowledged they might be missing the contribution and requested clarification, the detailed second-round rebuttal plus SG’23 comparison and stronger baselines could lift the score. A jump all the way to acceptance is unlikely without a fully satisfying SOTA comparison suite, but 2 to 4 is plausible if they accept the clarified novelty as “scene-level freehand sketch training + alignment” and accept code-availability constraints.

---

### Decision · Program_Chairs · 2026-01-26

Accept (Poster)